# A 41-year (1979-2019) passive microwave derived lake ice phenology data record of the Northern Hemisphere

Yu Cai[1], Claude R. Duguay[2,3], Chang-Qing Ke[1]

[1] Jiangsu Provincial Key Laboratory of Geographic Information Science and Technology, Key Laboratory for Land Satellite Remote Sensing Applications of Ministry of Natural Resources, School of Geography and Ocean Science, Nanjing University, Nanjing, China

[2] Department of Geography and Environmental Management, University of Waterloo, Ontario, Canada

[3] H2O Geomatics Inc., Waterloo, Ontario, Canada

*Correspondence to*: Chang-Qing Ke (kecq@nju.edu.cn)

**Abstract:** Seasonal ice cover is one of the important attributes of lakes in middle and high latitude regions. The annual freeze-up
and break-up dates and the durations of ice cover (i.e., lake ice phenology) are sensitive to the weather and climate, and hence can be used as an indicator of climate variability and change. In addition to optical, active microwave, and raw passive microwave data that can provide daily observations, the Calibrated Enhanced Resolution Brightness Temperature (CETB) dataset available from the National Snow and Ice Data Center (NSIDC) provides an alternate source of passive microwave brightness temperature ($T_B$) measurements for the determination of lake ice phenology on a 3.125 km grid. This study used Scanning Multi-channel Microwave
Radiometer (SMMR), Special Sensor Microwave Image (SSM/I) and Special Sensor Microwave Imager/Sounder (SSMIS) data from the CETB dataset to extract the ice phenology for 56 lakes across the Northern Hemisphere from 1979 to 2019. According to the differences in $T_B$ between lake ice and open water, a threshold algorithm based on the moving t test method was applied to determine the lake ice status for grids located at least 6.25 km away from the lake shore, and the ice phenology dates for each lake were then extracted. When ice phenology could be extracted from more than one satellite over overlapping periods, results from
the satellite offering the largest number of observations were prioritized. The lake ice phenology results showed strong agreement with an existing product derived from Advanced Microwave Scanning Radiometer for EOS (AMSR-E) and Advanced Microwave Scanning Radiometer 2 (AMSR-2) data (2002 to 2015), with mean absolute errors of ice dates ranging from 2 to 4 days. Compared to near-shore in-situ observations, the lake ice results, while different in terms of spatial coverage, still showed overall consistency. The produced lake ice record also displayed significant consistency when compared to a historical record of annual maximum ice
cover of the Laurentian Great Lakes of North America. From 1979 to 2019, the average complete freezing duration and ice cover duration for lakes forming a complete ice cover on an annual basis were 153 and 161 days, respectively. The lake ice phenology dataset – a new climate data record (CDR) – will provide valuable information to the user community about the changing ice cover of lakes in the last four decades. The dataset is available at https://doi.org/10.1594/PANGAEA.937904 (Cai et al., 2021).

## 1 Introduction

Climate change is one of the major challenges facing humanity. New technologies and methods are urgently needed to monitor and quantify the rapid changes in climate at the regional and global scale. Lakes are closely tied to climate conditions and are characterized by many important parameters for long-term monitoring of climate change, including the coverage and duration of

lake ice. For lakes located at middle and high latitudes, the spatial and temporal coverage of lake ice and key phenological events provide important information about changes in weather and climate. Lake ice phenology describes the seasonal evolution of ice cover, including the freeze-up and break-up dates, and ice cover duration (Duguay et al., 2015; Sharma et al., 2016; Smejkalova et al., 2016). The presence or absence of lake ice affects lake-atmosphere interactions, thus affecting hydrological and ecological processes in lakes (Duguay et al., 2006; 2015; Mishra et al., 2011; Hampton et al., 2017; Knoll et al., 2019). The coverage and duration of lake ice also affect human activities such as transportation, fishing, and winter recreational activities (Brown and Duguay, 2010; Prowse et al., 2011; Du et al., 2017; Sharma et al., 2019). Changing climate conditions in the cold season will alter the temporal and spatial characteristics of mass (such as precipitation and suspended particles) and energy (such as solar radiation and atmospheric heat) input into the lake, thus affecting the freeze-thaw processes of lake ice (Mishra et al., 2011). On the other hand, changes in the timing of freeze-up and break-up will cause sudden changes in lake surface properties (such as albedo and roughness) and affect the exchange between lakes and the atmosphere.

Lake ice (as well as river ice) phenology is one of the most detailed climate data record (CDR) with the longest historical coverage. For example, the Global Lake and River Ice Phenology Database (GLRIPD) from the National Snow and Ice Data Center (NSIDC) contains ice phenology records for 865 sites, including 24,438 ice-on records and 33,370 ice-off records, where the earliest record can date back to 1443 (Benson et al., 2000). GLRIPD provides a valuable data source for the study of historical changes in river and lake ice in the Northern Hemisphere. Based on this dataset, several investigations have reported that the ice phenology of rivers and lakes in the Northern Hemisphere has been changing with trends towards later freeze-up and earlier break-up in different periods of the past century. For example, Magnuson et al. (2000) found that the freeze-up dates of river and lake ice had a delaying trend at a rate of 5.8 days/century, and the break-up dates had an advancing trend at a rate of 6.5 days/century from 1846 to 1995, and the change rates of freeze-up and break-up dates accelerated after 1950. Similarly, Benson et al. (2012) found that the freeze-up dates were occurring later at a rate of 0.03 – 0.16 day/year, and the break-up dates were occurring earlier at a rate of 0.05 – 0.19 day/year from 1855 to 2005, resulting in a shortening of ice cover duration at the rate of 0.07 – 0.43 day/year. Thus far, widespread lake ice loss has been found across the Northern Hemisphere and is expected to continue in the next decades (Weyhenmeyer et al., 2011; Magnuson and Lathrop, 2014; Sharma et al., 2019; Woolway and Merchant, 2019). Also, some lakes at high latitudes have changed from permanent/multi-year ice cover to seasonal ice cover, and some lakes in warm regions have changed from seasonal ice cover to intermittent ice cover (Surdu et al., 2016; Sharma et al., 2019). However, GLRIPD sites are mainly concentrated in North America, northern Europe, and Russia. There are large areas in the world where in-situ lake ice records are still lacking or incomplete (Weber et al., 2016). In addition, many sites (especially those in Canada and Russia) stopped recording freeze-up and break-up dates in the late 1980s and 1990s, resulting in a rapid decrease in the lake ice records in recent decades (Murfitt and Duguay, 2021).

Satellite remote sensing has the advantage of observing the Earth's surface over large areas at a fixed time interval and has been widely used to monitor lake ice in recent decades. Most spaceborne satellites can observe the status of lake ice based on the different signals returned by ice and water. According to the wavelength used, remote sensing of lake ice can be divided into two categories: optical remote sensing and microwave remote sensing. Optical sensors with medium resolution (ca. 250-1000 m) usually provide a short revisit period (daily) and therefore have been widely used to estimate lake ice phenology (Arp et al., 2013; Kropáček et al., 2013; Smejkalova et al., 2016; Weber et al., 2016). For example, the Advanced Very High Resolution Radiometer (AVHRR) reflectance data have been used to extend existing in-situ observations in Canada (Latifovic and Pouliot, 2007). The Moderate Resolution Imaging Spectroradiometer (MODIS) land surface temperature product (1 km) and snow cover product (500 m) have also been used to determine lake ice status and extract lake ice phenology (Nonaka et al., 2007; Hall et al., 2010; Pour et al., 2012; Weber et al., 2016; Kropáček et al., 2013; Cai et al., 2019; Wu et al., 2021). However, optical sensors are easily affected

by cloud cover and illumination conditions, which limits their ability for lake ice monitoring under cloudy weather and at high latitudes (Maslanik and Barry, 1987; Helfrich et al., 2007; Kang et al., 2012). Microwave remote sensing allows for acquisitions regardless of cloud cover and dust in the atmosphere and is not affected by illumination conditions (Engram et al., 2018; Geldsetzer et al., 2010). Active microwave provides capabilities for the monitoring lake ice based on the difference in backscatter between lake ice and open water. For example, the European Remote Sensing Satellite (ERS)-1/2 Synthetic Aperture Radar (SAR) (Jeffries et al., 1994; Morris et al., 1995; Duguay and Lafleur, 2003) and Radarsat-1/2 SAR (Duguay et al., 2002; Geldsetzer et al., 2010) have been successfully used to obtain lake ice status. However, existing active microwave technology is limited by the narrow swath width, relatively low temporal resolution (especially at lower latitudes), and short historical coverage, making it difficult to monitor lake ice daily at a large scale (Latifovic and Pouliot, 2007; Chaouch et al., 2014) as required by the Global Climate Observing System (GCOS; Belward et al., 2016). Passive microwave sensors can capture lake ice status based on the difference in microwave radiation emitted from lake ice and open water. Microwave radiometers mounted on existing polar-orbiting satellite platforms can provide daily observations across the Northern Hemisphere and therefore can be used to monitor lake ice phenology. For example, Du et al. (2017) used Advanced Microwave Scanning Radiometer for Earth Observing System and Advanced Microwave Scanning Radiometer 2 (AMSR-E/2) daily $T_B$ data to extract lake ice phenology in the Northern Hemisphere from 2002 to 2015. The Scanning Multi-channel Microwave Radiometer (SMMR), Special Sensor Microwave Image (SSM/I) and Special Sensor Microwave Imager/Sounder (SSMIS) data have also been used to monitor ice phenology for large lakes from 1979 to present (Ke et al., 2013; Cai et al., 2017; Su et al., 2021). However, the microwave signals naturally emitted by surface features are weak, and thus passive microwave data usually have a coarse spatial resolution (several km to tens of km), but they provide great advantages in the ice observations of oceans and large lakes (Chaouch et al., 2014; Dörnhöfer and Oppelt, 2016).

The latest Calibrated Enhanced Resolution Brightness Temperature (CETB) dataset released by NSIDC provides passive microwave $T_B$ data at multiple grid spacings from 25 km to up to 3.125 km, including SMMR, SSM/I, SSMIS, and AMSR-E data. The enhanced-resolution images were generated using the radiometer version of the Scatterometer image Reconstruction (rSIR) algorithm, which provides higher spatial resolution surface $T_B$ images with smaller total error compared with conventional drop-in-the-bucket gridded image formation (Long and Brodzik, 2016). Due to the coarse grid spacing (25 km) of the original SMMR, SSM/I & SSMIS data, previous studies used these data to extract the ice phenology for only 22 lakes (Su et al., 2021). The newly released dataset with finer grid spacing provides an alternative data source for lake ice phenology research in the Northern Hemisphere. In addition, different from the continuous data used in previous studies (which only contained $T_B$ data from Nimbus-7, F08, F11, F13 and F17, and the overlapping time between satellites were less than one year), the CETB dataset contains data from all satellite series (Nimbus-7, F08, F10, F11, F13, F14, F15, F16, F17, F18 and F19) with greater overlap and more abundant $T_B$ measurements.

This study uses SMMR, SSM/I & SSMIS data from the new CETB dataset to generate a lake ice phenology data record of the Northern Hemisphere from 1979 to 2019. Two problems need to be solved: 1) how to extract lake ice phenology events from SMMR, SSM/I & SSMIS data; and 2) how to select lake ice phenology results from multiple satellites with overlapping years. The workflow behind the production of the dataset of annual ice dates and durations for 56 lakes is described. Then, the accuracy of the derived ice dates is compared against existing in-situ observations and satellite-based lake ice datasets. Finally, the spatial characteristics of the lake ice phenology in the Northern Hemisphere are analyzed.

## 2 Data and methods

### 2.1 Selected lakes

To make sure that there is at least one complete passive microwave pixel in the center of the lake, lakes with a large enough area or a nearly circular shape were selected. As a result, 56 lakes were selected as the study lakes. Lake boundaries from the European Space Agency (ESA) Lakes Climate Change Initiative (Lakes_cci) project (https://climate.esa.int/en/projects/lakes/) were also obtained to determine distance of each pixel to lake shore as to avoid land contamination (see section 2.3.2 for details) (Figure 1, Table 1). Among the 56 lakes, 19 are in North America and another 27 lakes are in Eurasia. Ten lakes located in warmer regions, forming intermittent ice cover, were also contained in the study. Since lakes freeze-up in autumn or winter and break-up in spring or summer, in order to simplify the description of the study period, the lake ice period is determined to be from September to August of the following year. For example, 2019 refers to the period from September 2018 to August 2019. However, the generated dataset available to users also provides calendar dates.

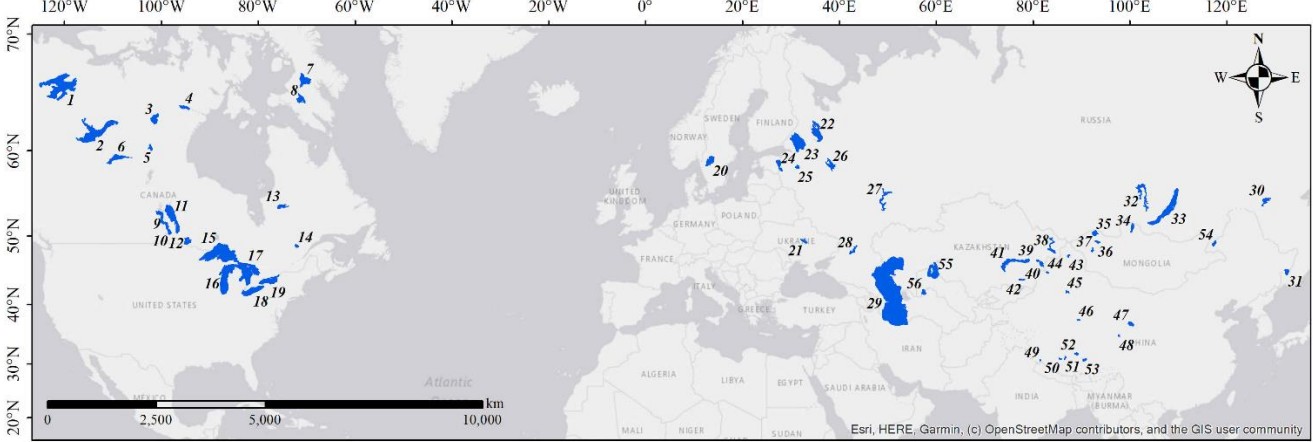

**Figure 1 Location of the 56 study lakes (the names of the lakes are listed in Table 1).**

**Table 1 The location and physical characteristics of the 56 study lakes, as well as the number of pixels used for the lake ice phenology extraction.**

| ID | Lake name | Country | Elevation (m) | Area (km²) | Average depth (m) | Number of pixels | Representativeness | Years |
|----|-----------|---------|---------------|------------|-------------------|------------------|--------------------|-------|
| 1 | Great Bear | Canada | 145 | 30450.6 | 72.2 | 1779 | 57.1% | 1979 ~ 2019 |
| 2 | Great Slave | Canada | 148 | 26734.3 | 59.1 | 1275 | 46.6% | 1979 ~ 2019 |
| 3 | Dubawnt | Canada | 218 | 3583.2 | 25.5 | 42 | 11.4% | 1979 ~ 2019 |
| 4 | Baker | Canada | 2 | 1664.7 | 60.0 | 12 | 7.0% | 1979 ~ 2019 |
| 5 | Kasba | Canada | 324 | 1330.1 | 15.7 | 3 | 2.2% | 1979 ~ 2019 |
| 6 | Athabasca | Canada | 207 | 7528.7 | 20.6 | 263 | 34.1% | 1979 ~ 2019 |
| 7 | Nettilling | Canada | 18 | 4872.7 | 23.4 | 103 | 20.6% | 1979 ~ 2019 |
| 8 | Amadjuak | Canada | 91 | 2994.9 | 24.8 | 81 | 26.4% | 1979 ~ 2019 |
| 9 | Winnipegosis | Canada | 251 | 5035.5 | 3.2 | 33 | 6.4% | 1979 ~ 2019 |
| 10 | Manitoba | Canada | 245 | 4751.1 | 3.6 | 120 | 24.7% | 1979 ~ 2019 |
| 11 | Winnipeg | Canada | 215 | 23923.0 | 11.9 | 1269 | 51.8% | 1979 ~ 2019 |
| 12 | Woods | Canada | 320 | 3472.8 | 10.7 | 29 | 8.2% | 1979 ~ 2019 |
| 13 | La Grande 3 Reservoir | Canada | 250 | 2401.0 | 25.0 | 1 | 0.4% | 1988 ~ 2019 |
| 14 | Saint-Jean | Canada | 96 | 1112.8 | 11.0 | 27 | 23.7% | 1988 ~ 2019 |
| 15 | Superior | Canada/USA | 179 | 81843.9 | 146.7 | 6438 | 76.8% | 1979 ~ 2019 |

| 16 | Michigan | USA | 175 | 57726.8 | 84.2 | 4364 | 73.8% | 1979 ~ 2019 |
|----|----------|-----|-----|---------|------|------|-------|-------------|
| 17 | Huron | Canada/USA | 175 | 59399.3 | 59.8 | 4129 | 67.9% | 1979 ~ 2019 |
| 18 | Erie | Canada/USA | 172 | 25767.8 | 19.4 | 1824 | 69.1% | 1979 ~ 2019 |
| 19 | Ontario | Canada/USA | 73 | 19347.4 | 84.8 | 1316 | 66.4% | 1988 ~ 2019 |
| 20 | Vanern | Sweden | 44 | 5486.2 | 27.9 | 165 | 29.4% | 1979 ~ 2019 |
| 21 | Kremenchuk Reservoir | Ukraine | 77 | 1849.1 | 7.3 | 10 | 5.3% | 1988 ~ 2019 |
| 22 | Onega | Russia | 13 | 9961.9 | 26.3 | 394 | 38.6% | 1979 ~ 2019 |
| 23 | Ladoga | Russia | -10 | 17444.0 | 48.0 | 1220 | 68.3% | 1979 ~ 2019 |
| 24 | Peipsi | Russia | 28 | 3489.0 | 7.2 | 134 | 37.5% | 1979 ~ 2019 |
| 25 | Ilmen | Russia | 16 | 959.8 | 12.5 | 20 | 20.4% | 1979 ~ 2019 |
| 26 | Rybinsk Reservoir | Russia | 97 | 4042.3 | 6.3 | 107 | 25.8% | 1979 ~ 2019 |
| 27 | Kuybyshev Reservoir | Russia | 45 | 5060.1 | 11.5 | 30 | 5.8% | 1988 ~ 2019 |
| 28 | Tsimlyanskoye Reservoir | Russia | 30 | 2253.9 | 10.6 | 13 | 5.6% | 1988 ~ 2019 |
| 29 | Caspian Sea | Russia | -29 | 377002.0 | 200.5 | 34148 | 88.5% | 1979 ~ 2019 |
| 30 | Zeyskoye Reservoir | Russia | 308 | 2234.7 | 30.6 | 20 | 8.7% | 1979 ~ 2019 |
| 31 | Khanka | Russia | 64 | 4118.8 | 4.4 | 246 | 58.3% | 1979 ~ 2019 |
| 32 | Bratsk Reservoir | Russia | 391 | 4810.7 | 35.1 | 6 | 1.2% | 1988 ~ 2019 |
| 33 | Baikal | Russia | 449 | 31967.8 | 738.7 | 1996 | 61.0% | 1979 ~ 2019 |
| 34 | Khovsgol | Mongolia | 1642 | 2767.8 | 138.6 | 73 | 25.8% | 1979 ~ 2019 |
| 35 | Uvs | Mongolia | 759 | 3600.8 | 9.9 | 201 | 54.5% | 1979 ~ 2019 |
| 36 | Khyargas | Mongolia | 1029 | 1383.2 | 54.4 | 17 | 12.0% | 1988 ~ 2019 |
| 37 | Khar Us | Mongolia | 1156 | 1120.5 | 2.8 | 8 | 7.0% | 1988 ~ 2019 |
| 38 | Zaysan | Kazakhstan | 388 | 4193.6 | 12.6 | 94 | 21.9% | 1988 ~ 2019 |
| 39 | Sasykkol | Kazakhstan | 348 | 744.8 | 3.2 | 2 | 2.6% | 1988 ~ 2019 |
| 40 | Alakol | Kazakhstan | 347 | 2919.3 | 22.1 | 98 | 32.8% | 1979 ~ 2019 |
| 41 | Balkhash | Kazakhstan | 338 | 16717.9 | 6.7 | 653 | 38.1% | 1979 ~ 2019 |
| 42 | Qapshaghay Bogeni Reservoir | Kazakhstan | 475 | 1206.0 | 23.3 | 6 | 4.9% | 1988 ~ 2019 |
| 43 | Ulungur | China | 478 | 854.9 | 8.0 | 16 | 18.3% | 1988 ~ 2019 |
| 44 | Ebi | China | 194 | 564.9 | 1.4 | 9 | 15.6% | 1988 ~ 2019 |
| 45 | Bosten | China | 1050 | 961.8 | 9.1 | 6 | 6.1% | 1988 ~ 2019 |
| 46 | Ayakkum | China | 3876 | 616.3 | 10.0 | 4 | 6.3% | 1988 ~ 2019 |
| 47 | Qinghai | China | 3194 | 4266.6 | 16.8 | 197 | 45.1% | 1979 ~ 2019 |
| 48 | Ngoring | China | 4267 | 617.8 | 17.4 | 2 | 3.2% | 1988 ~ 2019 |
| 49 | Ma-p'ang yung-ts'o | China | 4585 | 413.6 | 44.8 | 5 | 11.8% | 1988 ~ 2019 |
| 50 | Zhari Namco | China | 4612 | 958.1 | 25.0 | 9 | 9.2% | 1988 ~ 2019 |
| 51 | Tangra | China | 4535 | 825.1 | 120.0 | 1 | 1.2% | 1988 ~ 2019 |
| 52 | Siling | China | 4539 | 1749.5 | 28.0 | 44 | 24.6% | 1979 ~ 2019 |
| 53 | Nam | China | 4724 | 1963.8 | 44.4 | 53 | 26.4% | 1988 ~ 2019 |
| 54 | Hulun | China | 540 | 2121.4 | 6.2 | 69 | 31.8% | 1979 ~ 2019 |
| 55 | Large Aral Sea | Uzbekistan | 29 | 23865.9 | 1.1 | 1506 | 61.6% | 1979 ~ 2003 |
| 56 | Sarygamysh | Turkmenistan | 2 | 3772.0 | 12.2 | 214 | 55.4% | 1979 ~ 2019 |


## 2.2 Datasets

### 2.2.1 CETB dataset

The CETB dataset consists of gridded enhanced-resolution $T_B$ data produced from SMMR on Nimbus-7, SSM/I on Defense Meteorological Satellite Program (DMSP) 5D-2/F8, F10, F11, F13, F14, DMSP 5D-3/F15, SSMIS on DMSP 5D-3/F16, F17, F18,

F19, and AMSR-E on Aqua. For different channels, CETB provides $T_B$ data with multiple grid spacings from 25 km to 3.125 km. The enhanced-resolution images were generated using the rSIR algorithm where each pixel $T_B$ was derived from all overlapping measurements, weighted based on the antenna measurement response function (Brodzik et al., 2020). Compared to other existing passive microwave data products, CETB provides the most complete passive microwave records with the best spatial resolution and other advantages, such as improvements in cross-sensor calibration and quality checks, improved projection grids, and local

time-of-day processing (Brodzik et al., 2020). In this study, the 3.125 km 37 GHz H-polarized evening $T_B$ data from all the SMMR, SSM/I & SSMIS satellites (except for F19 since its temporal coverage was less than one year) were used to extract lake ice phenology in the Northern Hemisphere.

### 2.2.1 GLRIPD

GLRIPD contains in-situ observations of ice on and ice off dates for 865 lakes and rivers of the Northern Hemisphere. In this

database, ice on is defined as the freeze-up end date, and ice off represents the date of break-up end (Benson et al., 2000). For comparison with lake ice phenology extracted from SMMR, SSM/I & SSMIS data, ice on/off records were extracted for the sites that have at least ten years of overlap with study lakes (a lake may have multiple sites). Since the number of sites has decreased rapidly from the 1990s, records for only 11 sites on 7 lakes (Baikal, Baker, Great Slave, Onega, Superior, Winnipeg, and Ladoga) were used for comparison. It is worth noting that these are shore-based observations, so some differences in ice dates are expected

against satellite-based estimates that are determined from larger sections of the lakes.

### 2.2.2 AMSR-E/2 lake ice phenology product

The daily AMSR-E/2 lake ice phenology product (Du et al., 2017) was also used to compare with the lake ice phenology results derived from the CETB dataset. The product was produced based on 36.5 GHz H-polarized $T_B$ data, and all the data are spatially resampled to a 5-km grid. The temporal coverage of the AMSR-E ice product is from 4 June 2002 to 3 October 2011, and that for

AMSR2 is from 24 July 2012 to 31 December 2015. For each lake, a buffer of one-pixel (5 km in size) was set to remove the pixels near lake shore that may be contaminated by land. In Du et al. (2017), the lake ice phenology dates were obtained by using the thresholds of 5% and 95% of the total lake pixels (see section 2.3.2 for details).

### 2.2.3 GLERL ice cover

The Great Lakes Environmental Research Laboratory (GLERL) from the National Oceanic and Atmospheric Administration

(NOAA) provides historical ice cover data for the Laurentian Great Lakes of North America from 1973 to present. The data are available from the Canadian Ice Service (from 1973 to 1988) and the United States National Ice Center (1989 to present). The annual maximum ice cover records in this database were used for comparison with those extracted from SMMR, SSM/I & SSMIS data from 1979 to 2019.

### 2.3 Methods

### 2.3.1 Determination of lake ice status for each pixel based on MTT algorithm

Moving t test (MTT) is a method for detecting abrupt changes in time series by determining whether two sub-samples are significantly different from one another (Jiang and You, 1996; Xiao and Li, 2007). To obtain the lake ice phenology from 5 km 36.5 GHz H-polarized AMSR-E/2 data, Du et al. (2017) proposed a threshold algorithm based on MTT. In this study, we used the same algorithm to extract lake ice phenology from 3.125 km 37 GHz H-polarized SMMR, SSM/I & SSMIS data from 1979 to

2019. The algorithm consists of three steps (details on the processing chain and formulas are provided in Du et al. (2017)) and was

applied to each passive microwave pixel:

Step 1: For the $T_B$ of each pixel, MTT was used to detect all the sudden-change points in the time series.

Step 2: Based on the results of MTT, the reference $T_B$ of lake ice and open water were determined, and the threshold was

calculated by averaging the two reference $T_B$ .

Step 3: Based on the threshold, the lake ice status of the pixel for each day was determined (Figure 2a-b).

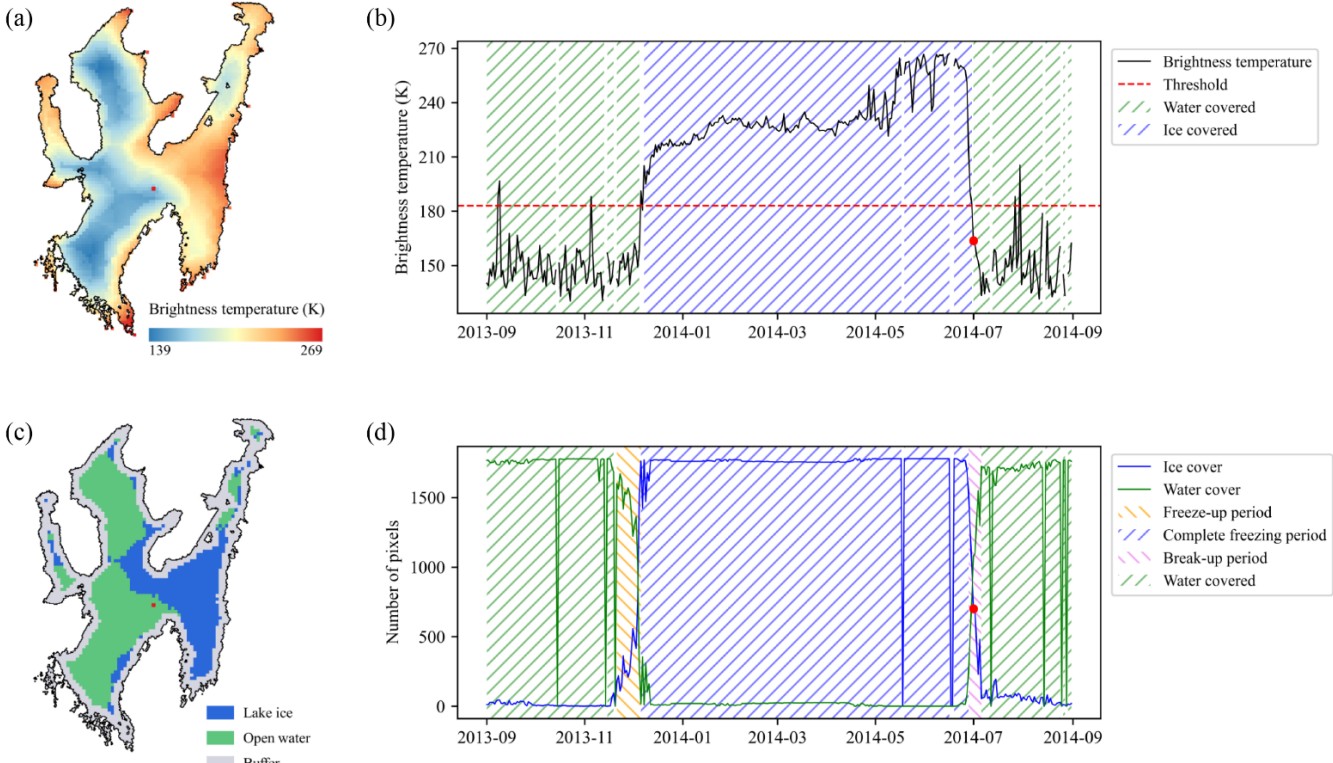

**Figure 2** Example of determining the lake ice status for a pixel and extracting the ice phenology for a lake. (a) The $T_B$ image of Great Bear Lake on 1 July 2014, (b) variations in the $T_B$ of the pixel marked in red in (a), the red line represents the reference $T_B$ determined by MTT algorithm, and the pixel was determined to be water covered on 1 July 2014 (red dot), (c) lake ice status for

all the pixels at least 6.25 km away from the lake shore on 1 July 2014, and (d) variations in the number of lake ice and open water pixels for Great Bear Lake in 2014, the lake ice phenology were extracted by thresholds of 5% and 95% of the total lake pixels, and 1 July 2014 (red dot on the ice cover line) was determined to be in break-up period.

### 2.3.2 Extraction of lake ice phenology for each lake

For each pixel in a lake, the algorithm was applied for the $T_B$ series from each satellite. Afterwards, we set a buffer of two-pixel

size (6.25 km) from lake shore to exclude pixels that could be contaminated by a high land fraction. The pixel size of the CETB product (3.125 km) is smaller than the footprint of the original passive microwave data (ca. 30 km), which may result in mixed water (or ice) and land in pixels near the lake shore. When the lake is water covered, the $T_B$ for land-contaminated pixels will be higher than that of a pure water pixel, while when the lake is ice covered, the $T_B$ will be lower than that of a pure ice pixel. As a result, for land-contaminated pixels, the $T_B$ difference between lake ice and water will be smaller than expected, which will

eventually affect the accuracy of MTT algorithm. Therefore, setting a buffer is necessary to exclude land-contaminated pixels. In addition, since passive microwave data have periodic sampling intervals at middle and low latitudes due to the orbital spacing,

linear interpolation was used to fill the gaps in the $T_B$ series before applying the MTT algorithm, but the interpolated data will not be used in the process of lake ice phenology extraction.

For the remaining pixels after filtering, the number of daily lake ice/lake water pixels were counted. Afterwards, thresholds of 5% and 95% of the total lake pixels were set to extract the lake ice phenology (Figure 2c-d). In fall or winter, when the number of lake ice pixels is larger than 5% of the total pixels, the lake is considered to start to form ice cover (freeze-up start), and if the number of lake ice pixels is larger than 95%, the lake is considered as completely frozen (freeze-up end). Similarly, in spring or summer, when the number of lake ice pixels is less than 95%, the lake ice is considered to start to break up (break-up start), and if the number of lake ice pixels is less than 5%, the lake is considered as completely ice-free (break-up end). Then, the ice duration can be calculated: the complete freezing duration represents the period from freeze-up end to break-up start; the ice cover duration represents the period from freeze-up start to break-up end.

Some lakes in warmer regions did not form ice cover in certain years. For these lakes with intermittent ice cover, if no ice cover was detected throughout the year, the ice cover duration was recorded as 0; if lake ice formed but did not completely cover the lake surface, the freeze-up start and break-up end dates were extracted, while the complete freezing duration was recorded as 0. To avoid the impact of short-time weather events on the $T_B$, only lake ice cover persisting for more than 30 consecutive days was recorded, otherwise, it was considered that there was no ice cover detected in the year.

### 2.3.3 Selection of lake ice phenology results from multiple passive microwave satellites

Since there are overlapping time periods for certain satellites, some years may have multiple lake ice results. Take Great Bear Lake as an example, 103-year lake ice phenology results can be obtained from all ten satellites. However, these data were not evenly distributed within the 41 years. The lake ice phenology can be obtained simultaneously from four satellites from 2011 to 2019, while only data from one satellite can be used from 1979 to 1992 (Figure 3).

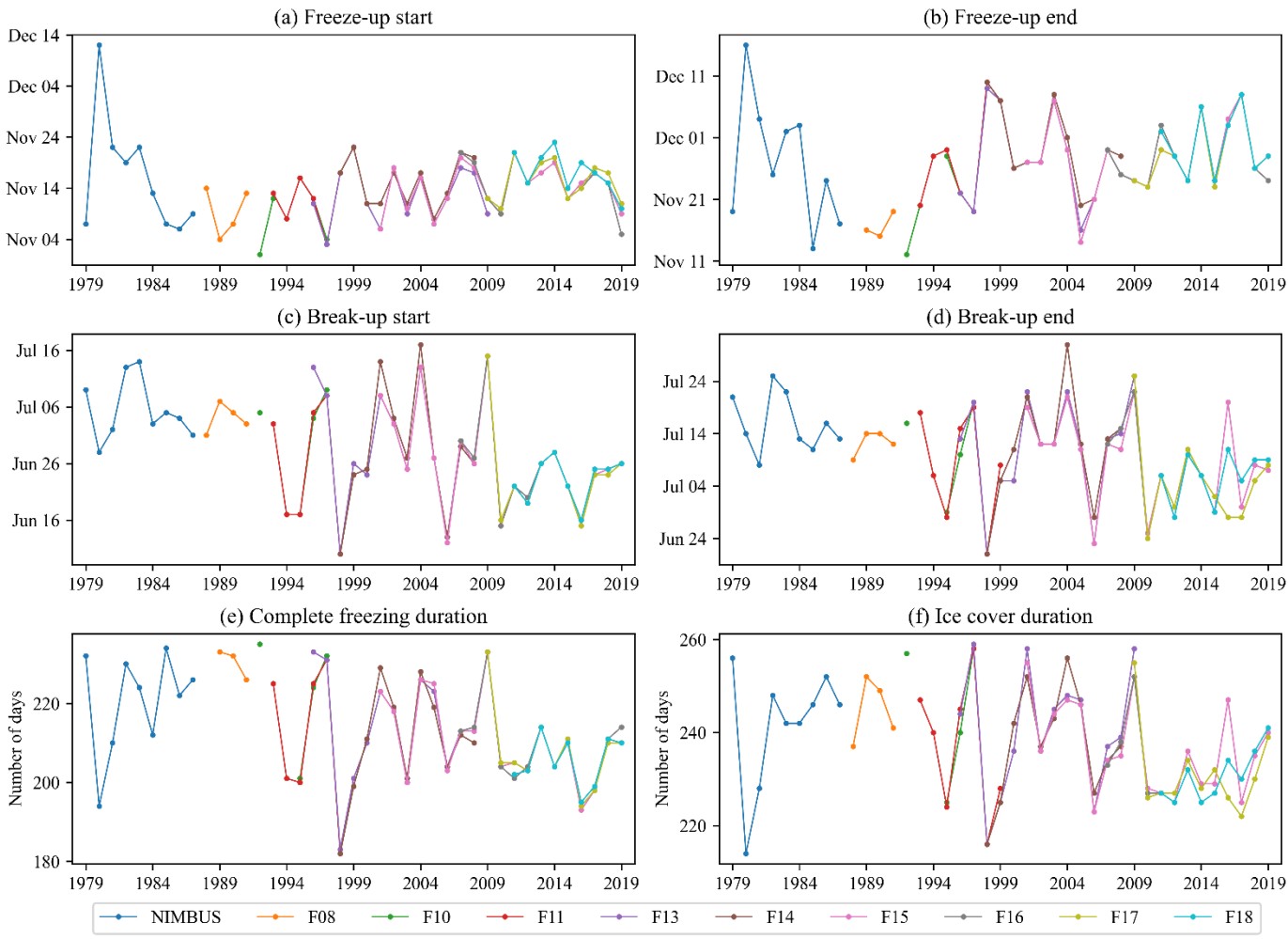

**Figure 3 Lake ice phenology for Great Bear Lake extracted from multiple satellites. (a) Freeze-up start date, (b) freeze-up end date, (c) break-up start date, (d) break-up end date, (e) complete freezing duration, and (f) ice cover duration.**

215       For each satellite, only data covering a complete year (from September to August of the following year) were retained for lake ice phenology extraction, except for Nimbus-7 data (with no overlapping data from other satellites), which acquired data from 25 October 1978 until 20 August 1987. Therefore, the freeze-up dates of 1987 could not be obtained for some lakes. In addition, an abnormal $T_B$ increase in April 1986 can be observed for some lakes, which may be due to the special operations period from 2 April to 23 June 1986. To avoid possible mistakes, the break-up dates for these lakes were not extracted.

220       Although all the satellites have a nominal daily temporal resolution, they have different frequencies of missing retrievals due to the polar orbit pattern and different sensor swath width. Since there are no alternative data for Nimbus-7 and F08, the lake ice phenology for some lakes prior to 1992 (especially before 1987 due to the low temporal resolution and poor data quality of SMMR data) were not complete. After the launch of F11, there are at least two satellites that can retrieve $T_B$ at the same time, which greatly improved the lake ice phenology results. As a result, lake ice phenology could be extracted for 34 lakes over 41 years (starting in 225   1979, Nimbus-7 data), and for another 21 lakes for more than 30 years (starting in 1988, F08 data). In addition, Large Aral Sea experienced a remarkable decrease in its extent during the study period, the increase of land pixels seriously affected the extraction of ice dates, hence ice phenology was only extracted from 1979 to 2003.

      Ideally, we can obtain four ice dates (freeze-up start, freeze-up end, break-up start, and break-up end dates) in a year. However, due to data quality limitation, we failed to obtain effective observations for some lakes in some years. Take F08 as an example, for 230   all the 56 lakes from 1988 to 1991, we should have obtained 896 records (four ice dates for each lake in each year), but mainly due

to the continuous missing data in the winter of 1987, we finally obtained 842 ice dates. Based on the number of observations that were actually available to those that were expected, we calculated the percentage of effective observations (which was 93.97% for F08, Table 2), as the basis for selecting annual ice phenology dates from overlapping results. With more missing data and a larger footprint, SMMR from the Nimbus satellite had the lowest percentage of effective observations among all the ten satellites.

Unfortunately, there were no alternative data to improve the poor data results caused by the data quality. SSM/I data from the six satellites all can attain more than 90% effective observations. Unexpectedly, we did not obtain the best lake ice results from the three SSMIS satellites, their effective observation percentage were all lower than 90%.

We ranked the percentage of effective observations of the different satellites in a priority list for obtaining lake ice phenology (Table 2). For years with multiple lake ice records extracted from more than one satellite, we prioritized the results from the satellite

with the highest percentage of effective observations. Therefore, if a lake had complete lake ice results from all the satellites, we would use the results from Nimbus-7 for 1979 to 1987, F08 for 1988 to 1991, F10 for 1992 to 1995, F13 for 1996 to 1997 and 2009, F14 for 1998 to 2008, and F15 for 2010 to 2019.

**Table 2 Percentage of effective observations and the sequence for different satellites**

| Satellite | Sensor | Percentage of effective observations | Priority | Temporal coverage | Study period |
|---|---|---|---|---|---|
| Nimbus | SMMR | 47.12% | 10 | 1978/10/25 - 1987/08/20 | 1979 - 1987 |
| F08 | SSM/I | 93.97% | 5 | 1987/07/09 - 1991/12/31 | 1988 - 1991 |
| F10 | SSM/I | 95.39% | 3 | 1990/12/08 - 1997/11/14 | 1992 - 1997 |
| F11 | SSM/I | 91.07% | 6 | 1991/12/03 - 2000/05/16 | 1993 - 1999 |
| F13 | SSM/I | 95.41% | 2 | 1995/05/03 - 2009/11/19 | 1996 - 2009 |
| F14 | SSM/I | 96.55% | 1 | 1997/05/07 - 2008/08/23 | 1998 - 2008 |
| F15 | SSM/I | 94.15% | 4 | 2000/02/23 - 2019/12/31 | 2001 - 2019 |
| F16 | SSMIS | 84.82% | 8 | 2005/11/01 - 2019/12/31 | 2007 - 2019 |
| F17 | SSMIS | 87.38% | 7 | 2008/03/01 - 2019/12/31 | 2009 - 2019 |
| F18 | SSMIS | 83.88% | 9 | 2010/03/08 - 2019/12/31 | 2011 - 2019 |

Finally, a data record of annual ice phenology for the 56 lakes from 1979 to 2019 was obtained. To increase the completeness of the lake ice phenology records, break-up dates for two lakes (Nettilling and Amadjuak) in 1987 were obtained from early F08 data (the two lakes had ice cover until late August, while Nimbus-7 data ended on 20 August 1987, and F08 started from 9 July 1987). Apart from the automatically extracted lake ice dates, 376 lake ice dates (i.e., 2.08% of all the records) were manually extracted to increase the ice phenology records (by setting a looser threshold in the extraction of lake ice phenology dates). Among

these manually extracted lake ice dates, 263 dates were extracted from Nimbus-7 (i.e., 24.24% of all the Nimbus-7 records).

## 3 Results and Discussions

### 3.1 Uncertainties of passive microwave derived lake ice phenology

There are two main error sources of lake ice phenology derived from passive microwave data: 1) the periodically missing data caused by the polar orbit operation mode of passive microwave satellites; and 2) errors associated with the extraction process of

lake ice phenology. Although the nominal temporal resolution of passive microwave data was one day, there were periodical gaps for $T_B$ data due to the existence of orbital spacing. Overall, the sampling frequency for SMMR data was the lowest, which may cause a maximum error of 4 – 6 days in the process of lake ice phenology extraction for lakes at low latitudes from 1979 to 1987.

The sampling frequency greatly increased with SSM/I and SSMIS, such that the potential error caused by data gaps was 1 – 3 days from 1988 to 2019 (Cai et al., 2017). If missing data was encountered in the extraction process of lake ice phenology, we would obtain the next valid date as the result of lake ice phenology, so the uncertainties caused by missing data were all negative values. The average uncertainty of all records in the dataset were -1 day, of which 61.70% of the records were not affected by missing data. The lower frequency of SMMR data led to larger uncertainty (average -3 days) in the lake ice phenology results, which decreased significantly after 1988 (Figure 4a). For each satellite, the frequencies of missing data in different regions were different: the lower the latitude, the higher the frequency of missing data. For example, there were 2 days of missing data every 4 – 5 days for F15, but the impact of sampling intervals on lakes at low latitudes (such as Qinghai Lake, at 36.9°N) was greater than that of lakes at middle latitudes (such as Baikal Lake, at 52.2°N), while lakes at high latitudes would not be affected by the sampling intervals (such as Great Bear Lake, at 65.1°N) (Figure 5). Figure 4b shows the absolute average uncertainty for each lake ordered by latitude (records extracted from SMMR data were not included because not all lakes had records from 1979 to 1987), it can be seen that the average uncertainty is larger for lakes found at latitudes below 40°N (Ayakkum, Qinghai, Ngoring, Siling, Tangra, Zhari Namco, Nam, and Ma-p'ang yung-ts'o, all on the Tibetan Plateau). In addition, the uncertainty describes the extreme error that may exist in lake ice phenology dates due to missing data, while the actual error is more likely to be smaller than the uncertainty. For example, the F08 data had 43 consecutive days of missing data from 1 December 1987 to 12 January 1988, while Lake Ladoga started to form ice cover during the period and had an ice coverage of 34% on 13 January 1988. As a result, we recorded 13 January 1988 as the freeze-up start date of Lake Ladoga with an uncertainty of -43 days (one of the largest uncertainties), but its actual error was probably much smaller than 43 days.

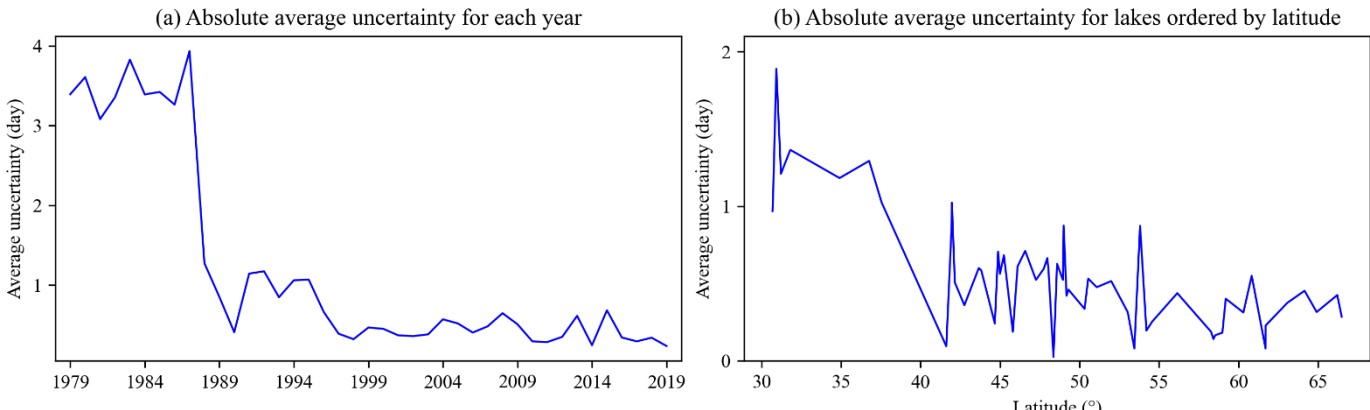

**Figure 4 Differences in absolute average uncertainty among years and lakes. (a) Absolute average uncertainty for each year; and (b) absolute average uncertainty of lake ice phenology results extracted from SSM/I and SSMIS data for each lake ordered by latitude.**

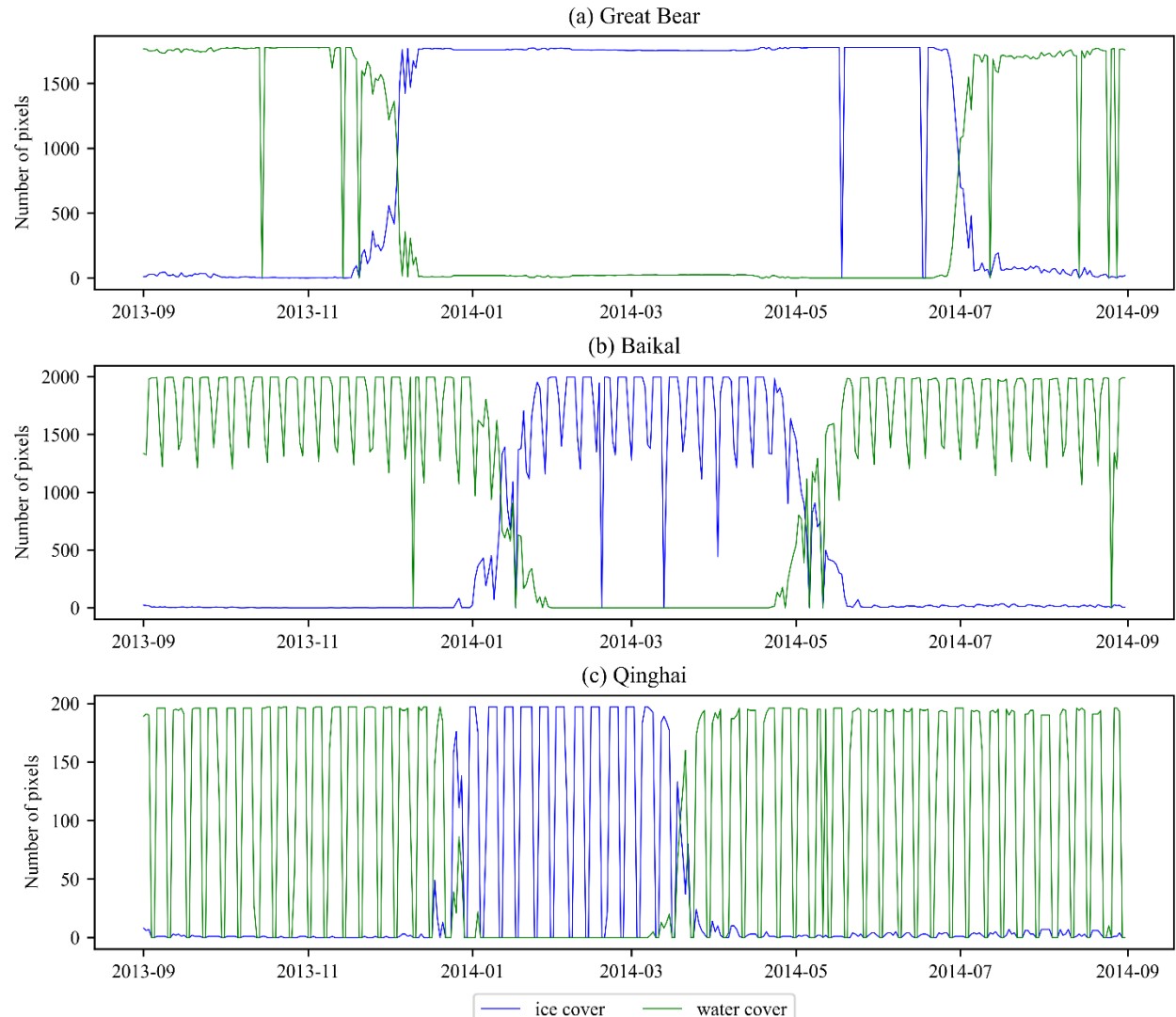

**Figure 5 Variations in the number of lake ice and open water pixels for (a) Great Bear, (b) Baikal, and (c) Qinghai in 2014. The periodical fluctuations in the curves are due to the existence of orbital spacing.**

The errors in the extraction of lake ice phenology were mainly caused by mixed pixels. Although the grid spacing for the enhanced resolution $T_B$ data was 3.125 km, the footprint of original data was approximately 30 km. Therefore, a single pixel may contain $T_B$ information from a variety of different types of surface features (especially for pixels near lake shore) (Bellerby et al., 1998; Bennartz, 2010). During the process of freeze-up/break-up, a pure pixel may experience a $T_B$ variation of up to 70 – 140 K (Kang et al., 2012), but the influence from land composition will result in a sharp decrease in the difference of $T_B$ between lake ice and open water. Since the MTT algorithm was mainly based on the sudden changes in the $T_B$ series, the mixed pixels will directly affect the ice status determined by the algorithm. Before extracting the lake ice phenology dates, we set a buffer of two pixels (6.25 km) to exclude pixels near lake shore. This is a compromise between removing more mixed pixels and extracting ice phenology for more lakes. However, the setting of a buffer will cause the loss of $T_B$ information near lake shore. Based on the number of pixels we used and the lake area, we calculated the representativeness of the pixels for each lake (Table 1). Depending on the lake area and shore complexity, and the possible existence of islands on the lake, the representativeness of the pixels ranged from 0.4% (La Grande 3 Reservoir) to 88.5% (Caspian Sea). For lakes with a low representativeness, the setting of the buffer may result in a non-negligible error in the lake ice phenology results which is hard to quantify. Since the freeze-up and break-up of ice cover

usually starts from lake shore (especially for the freeze-up), the beginning signals of freeze-up and break-up extracted from the retained pixels may be later compared with the actual ice conditions, while the ending signals may be earlier.

In addition, for each pixel, we only determined whether the pixel was covered by lake ice or not. However, the area for a single passive microwave pixel is 9.77 km$^2$, and the water (/ice) within this area may not freeze-up (/break-up) at the same time. Before the $T_B$ exceeds (/falls below) the threshold, the lake surface within the pixel may have already started to freeze-up (/break-up), and this process may not end even after we detect the ice covered (/water covered) signal. When the lake area is large enough, the gradual freeze-up or break-up within the pixel can be ignored, but for lakes with few available pixels and low representativeness, it may lead to certain deviations in the lake ice phenology results similar to the effect of setting a buffer.

Furthermore, during application of the MTT algorithm, multiple smoothing approaches were applied to the original $T_B$ series to exclude potentially dynamic $T_B$ changes caused by short-term weather events. For example, when using MTT to judge whether there was a sudden $T_B$ change on a certain day, the $T_B$ of 20 days before and after that day were averaged to make the comparison (Du et al., 2017). As a result, the MTT algorithm may be insensitive to short-term ice cover or frequent melt-refreeze event during the break-up or freeze-up seasons. 37 GHz H-polarized is sensitive to wind-induced surface roughness during the open water period (Kang et al., 2012), an affect that can also be attenuated by smoothing approaches, but may still lead to errors in lake ice phenology results. In addition, we only kept the lake ice phenology results with ice cover persisting for more than 30 consecutive days, which may also lead to some lakes with short-term ice cover being recorded as ice-free. Du et al. (2017) also pointed out that relatively small $T_B$ increases caused by thin ice may not be detectable using the MTT algorithm. Overall, the limitations in data and algorithm may lead to an underestimation of the ice cover duration, especially for small lakes with irregular shape and short-term ice cover.

Therefore, we adopted an automatic algorithm to extract lake ice phenology dates from all the platforms for all the lakes and prioritized the lake ice phenology results extracted from the satellite with higher percentage of effective observations, which was in order to ensure the comparability of lake ice phenology results among lakes and the consistency in time series. Despite the uncertainties and inevitable errors caused by the periodically missing data and mixed pixels, the lake ice phenology derived from SMMR, SSM/I & SSMIS can provide reliable information about the differences among lakes and the variations in time series.

## 3.2 Comparisons of lake ice phenology results from multiple satellites

For all the satellite pairs with overlapping years, we calculated the bias and mean absolute error (MAE) of the overlapping results. Different satellites can obtain lake ice dates with a bias from -1 to 2 days and an MAE from 1 to 3 days. The freeze-up end and break-up start dates extracted from different satellites had relatively higher consistencies than freeze-up start and break-up end dates (Table 3). The same conclusion can be drawn from the example of Great Bear Lake (Figure 3). The differences among the satellites were the reason why we decided to use a priority list to determine the lake ice phenology from overlapping results instead of calculating their average values or choosing the earliest freeze-up dates and latest break-up dates. The lake ice phenology determined by the satellite priority can be more consistent and comparable among different years and different lakes, which is more beneficial for analyzing their spatial differences and change trends.

**Table 3 Comparisons of lake ice phenology results from different satellites. FUS, FUE, BUS, and BUE represent freeze-up start, freeze-up end, break-up start, and break-up end date, respectively.**

| Compared satellites | Compared sensors | Bias | | | | MAE | | | | Overlapping years |
|---|---|---|---|---|---|---|---|---|---|---|
| | | FUS | FUE | BUS | BUE | FUS | FUE | BUS | BUE | |
| F10 - F11 | SSM/I - SSM/I | 0 | 0 | -1 | -1 | 3 | 2 | 2 | 3 | 1993 ~ 1997 |
| F10 - F13 | | 0 | 0 | 0 | 0 | 3 | 2 | 2 | 2 | 1996 ~ 1997 |
| F11 - F13 | | -1 | 0 | 0 | 1 | 3 | 2 | 2 | 2 | 1996 ~ 1999 |
| F11 - F14 | | 0 | 0 | 0 | 1 | 2 | 2 | 2 | 2 | 1998 ~ 1999 |
| F13 - F14 | | 0 | 0 | 0 | 0 | 2 | 2 | 1 | 2 | 1998 ~ 2008 |
| F13 - F15 | | 0 | 0 | 0 | 1 | 2 | 2 | 1 | 2 | 2001 ~ 2009 |
| F14 - F15 | | 0 | 0 | 0 | 1 | 2 | 2 | 1 | 1 | 2001 ~ 2008 |
| F13 - F16 | SSM/I - SSMIS | 0 | 0 | 0 | 1 | 3 | 2 | 2 | 2 | 2007 ~ 2009 |
| F13 - F17 | | 1 | 0 | 0 | 1 | 3 | 1 | 2 | 2 | 2009 |
| F14 - F16 | | -1 | 0 | 1 | 1 | 3 | 2 | 2 | 2 | 2007 ~ 2008 |
| F15 - F16 | | -1 | 0 | 0 | 0 | 2 | 2 | 2 | 2 | 2007 ~ 2019 |
| F15 - F17 | | -1 | 0 | 0 | 1 | 2 | 2 | 2 | 2 | 2009 ~ 2019 |
| F15 - F18 | | -1 | -1 | 1 | 2 | 2 | 2 | 2 | 2 | 2011 ~ 2019 |
| F16 - F17 | SSMIS - SSMIS | 0 | 0 | 0 | 1 | 2 | 2 | 2 | 2 | 2009 ~ 2019 |
| F16 - F18 | | -1 | -1 | 1 | 2 | 2 | 2 | 2 | 3 | 2011 ~ 2019 |
| F17 - F18 | | 0 | -1 | 0 | 0 | 2 | 2 | 2 | 2 | 2011 ~ 2019 |

## 3.3 Comparisons with existing lake ice datasets

### 3.3.1 Comparisons with in-situ observations

The lake ice dates recorded in the GLRIPD were compared with the results extracted from SMMR, SSM/I & SSMIS data. Since the ice on and ice off dates in the GLRIPD represent the first date of complete ice cover and the last date of break-up process, respectively, we used the freeze-up end and break-up end results to make the comparison. For Lake Superior and Ladoga with incomplete freeze-up end records, freeze-up start dates were used instead. Only the lake ice records with an overlapping time of more than 10 years were selected for the comparison. As a result, ice on dates for 5 lakes (8 sites) and ice off dates for 7 lakes (10

sites) were compared, and the correlation coefficient (r), mean difference (bias), and mean absolute error (MAE) were calculated (Table 4). It can be seen that the ice-on dates for 4 out of 8 sites, and the ice off dates for 8 out of 10 sites from GLRIPD are significantly consistent with the lake ice dates from SMMR, SSM/I & SSMIS data (Table 4). The main reason for the difference between lake ice dates from in-situ observations and passive microwave is tentatively attributed to their different observation ranges. In-situ records rely on observations of lake ice status visible from lake shores by human observers, while passive microwave

satellites record $T_B$ from the entire lake surface (here within the pre-defined buffer), but a follow-up investigation is indeed needed to quantify differences between in-situ observations with satellite-derived time series. Since break-up usually occurs more rapidly than freeze-up (Livingstone, 1997), the ice off dates from the two datasets were more consistent than the ice on dates (Table 4). With the decrease in in-situ observation sites globally since the 1980s, there has been a shift towards the increased use of remote sensing technology for lake ice monitoring (Murfitt and Duguay, 2021). Satellite observations can complement if not supersede

in-situ observations to investigate inter-annual variations and trends in lake ice phenology in recent decades.

**Table 4 Comparison of ice on/off dates from GLRIPD records and SMMR, SSM/I & SSMIS data. Bold number corresponds to p < 0.05.**

| Lake name | Site name | r | | Bias | | MAE | | Overlapping years | |
|---|---|---|---|---|---|---|---|---|---|
| | | ice on | ice off | ice on | ice off | ice on | ice off | ice on | ice off |
| Baikal | Lake Baikal | **0.68** | **0.65** | -4 | 31 | 6 | 31 | 1979 ~ 2006 | 1979 ~ 2006 |
| Baker | Baker Lake | | -0.05 | | 10 | | 10 | | 1979 ~ 1990 |
| Great Slave | Back Bay | 0.29 | **0.90** | 35 | 15 | 35 | 15 | 1979 ~ 1995 | 1979 ~ 1994 |
| | Great Slave Lake - Mcleod Bay | 0.31 | | 13 | | 13 | | 1979 ~ 1991 | |
| | Great Slave Lake - Charlton Bay | 0.33 | **0.89** | 16 | -7 | 16 | 7 | 1979 ~ 1991 | 1979 ~ 1990 |
| Onega | Lake Onega (Petrozavodsk) | | 0.55 | | 18 | | 18 | | 1979 ~ 1988 |
| | Lake Onega (Longasy) | | **0.79** | | 16 | | 16 | | 1979 ~ 1988 |
| Superior | Lake Superior at Bayfield | **0.53** | **0.65** | 24 | 6 | 24 | 12 | 1979 ~ 2009 | 1979 ~ 2009 |
| | Lake Superior at Madeline Island | **0.49** | **0.65** | 23 | 6 | 23 | 12 | 1979 ~ 2004 | 1979 ~ 2004 |
| Winnipeg | Lake Winnipeg | **0.75** | **0.85** | 3 | 14 | 5 | 14 | 1979 ~ 1991 | 1979 ~ 1990 |
| Ladoga | Lake Ladoga | 0.44 | **0.83** | -35 | 21 | 35 | 21 | 1979 ~ 1988 | 1979 ~ 1988 |

### 3.3.2 Comparisons with AMSR-E/2 lake ice phenology product

The AMSR-E/2 lake ice phenology product (Du et al., 2017) was also used to extract the ice dates for the study lakes from 2003 to 2015 (except for 2012 due to the data gap between AMSR-E and AMSR2 from 4 October 2011 to 23 July 2012), and the r, bias, and MAE values compared with the ice dates from SMMR, SSM/I & SSMIS data were calculated for each lake (Figure 6). The freeze-up start and break-up end dates for 55 lakes (except for Large Aral Sea with no records after 2003) and freeze-up end and break-up start dates from 49 lakes (except for Large Aral Sea and some lakes without complete ice cover in winter) were compared.

The break-up dates had an overall higher consistency between the results from the two datasets than the freeze-up dates (Figure 6a). Among them, the freeze-up start dates for 49 lakes, the freeze-up end dates for 45 lakes, the break-up start dates for 46 lakes, and the break-up end dates for 53 lakes were significantly consistent. The biases for all pairs of freeze-up start, freeze-up end, break-up start, and break-up end date were 2, -2, 3, and 0 days, respectively (Figure 6b). The freeze-up start and break-up start dates obtained from SMMR, SSM/I & SSMIS data were approximately 2 ~ 3 days later than the results from AMSR-E/2 data,

while the freeze-up end and break-up end dates were slightly earlier than those from AMSR-E/2 data. Although SMMR, SSM/I & SSMIS from CETB dataset have a grid spacing of 3.125 km, their original resolution was ca. 30 km, while AMSR-E/2 data (5-km grid spacing) have a finer original spatial resolution of about 10 km (14 × 8 km for AMSR-E and 12 × 7 km for AMSR-2). The buffers used to extract lake ice phenology dates were also different: the buffer set for SMMR, SSM/I & SSMIS data was 6.25 km and that for AMSR-E/2 data was 5 km. Therefore, AMSR-E/2 data captured more information near the lake shore than SMMR,

SSM/I & SSMIS data (the representativeness of AMSR-E/2 data ranged from 1.9% to 89.8%, with an average of 5.0% higher than that of SMMR, SSM/I & SSMIS data), which led to the directional differences between the lake ice phenology dates extracted from the two datasets. The MAEs for all the pairs of freeze-up start, freeze-up end, break-up start, and break-up end date were 4,

3, 3, and 2 days, respectively (Figure 6c). Among the four lake ice dates, the MAE of the freeze-up start dates was the largest, while that of the break-up end dates was the smallest. This is also due to the fact that the lake can take a longer time to freeze-up in early winter, but after the lake starts to break-up in spring, the lake surface will be completely ice-free in a short time (Livingstone, 1997). Overall, despite the different spatial resolution and sensor characteristics, the lake ice phenology results obtained from the two passive microwave data are highly consistent.

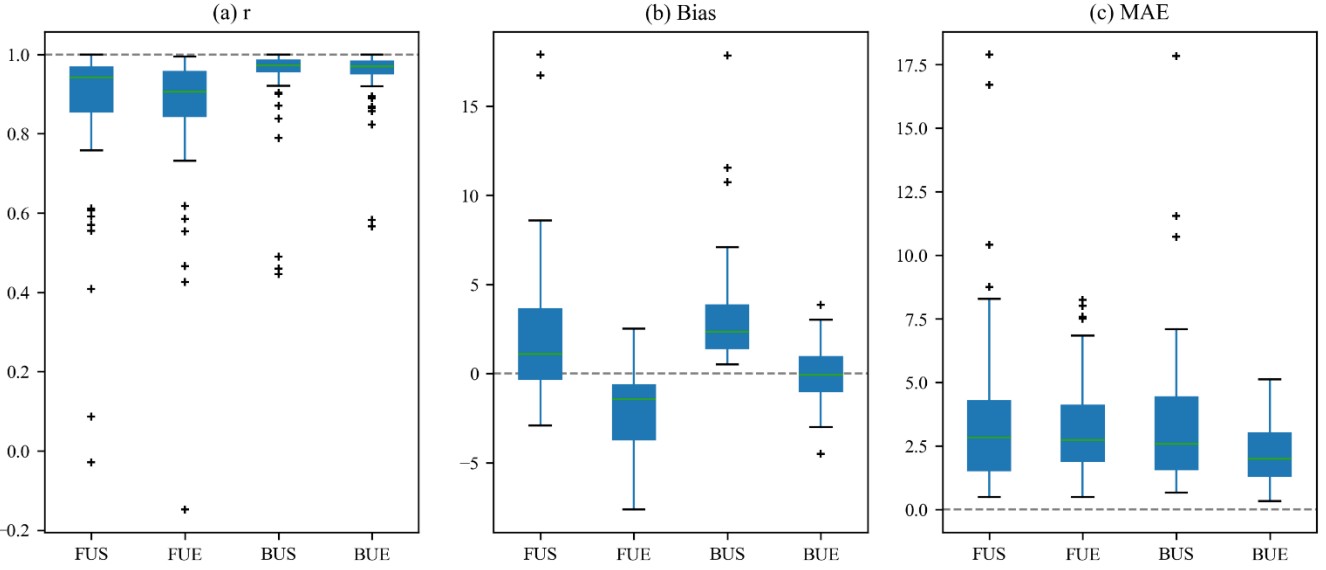

**Figure 6 Box plot of the (a) r, (b) bias, (c) mean absolute error (MAE) results of the comparison between the lake ice phenology dates extracted from AMSR-E/2 product and SMMR, SSM/I & SSMIS data.**

### 3.3.3 Comparisons with GLERL ice cover records

Apart from lake ice phenology dates, the annual maximum ice cover was extracted for seven intermittently ice-covered lakes with more than 1000 pixels (Superior, Huron, Michigan, Erie, Ontario, Caspian Sea, and Ladoga) from SMMR, SMM/I & SMMIS data and compared to that from the five Great Lakes contained in GLERL historical ice cover records. Ice cover maximums from SMMR, SSM/I & SSMIS data and GLERL were significantly consistent for these lakes (Table 5). Among them, the ice cover extracted for Erie showed the lowest bias and MAE, due to the fact that this lake experiences an extensive ice coverage in winter (Figure 7d). The remaining four lakes all showed a negative bias, indicating that the ice cover extracted from SMMR, SSM/I & SSMIS data were usually smaller than the actual situation. This is not only because a buffer of 6.25 km was used to exclude pixels near the lake shore, which happens to be the place where lake ice forms first, but short-term ice cover, which was common on these lakes, was difficult to be detected by MTT algorithm. This is also why sometimes the lake has been 100% ice covered but only partial coverage was detected by SMMR, SSM/I & SSMIS data. Taking 2014 as an example, we recalculated the daily ice cover from GLERL ice charts over the same areas as the SMMR, SSM/I & SSMIS data to make the comparison of daily ice cover changes from the two datasets (Figure 8). High consistency of daily ice cover percentage can be seen for Erie, Huron, and Superior. However, for Michigan and Ontario, a similar change pattern but lower ice cover was obtained from SMMR, SSM/I & SSMIS data compared to GLERL records because the short-term ice cover on Michigan and Ontario in February and March failed to be detected by MTT algorithm after the smoothing approaches was applied (Figure 8). Although it is difficult to capture the short-term changes in ice cover, the algorithm showed good performance of obtaining ice cover for lakes with long-term ice cover.

405 **Table 5 Comparison of annual maximum ice cover of the Great Lakes from GLERL records and SMMR, SSM/I & SSMIS data. Bold number corresponds to p < 0.05.**

| Lake | r | bias | MAE |
|---|---|---|---|
| Superior | **0.93** | -13.94 | 15.96 |
| Huron | **0.94** | -16.07 | 16.89 |
| Michigan | **0.79** | -23.77 | 23.77 |
| Erie | **0.91** | 0.65 | 7.32 |
| Ontario | **0.70** | -19.61 | 19.61 |

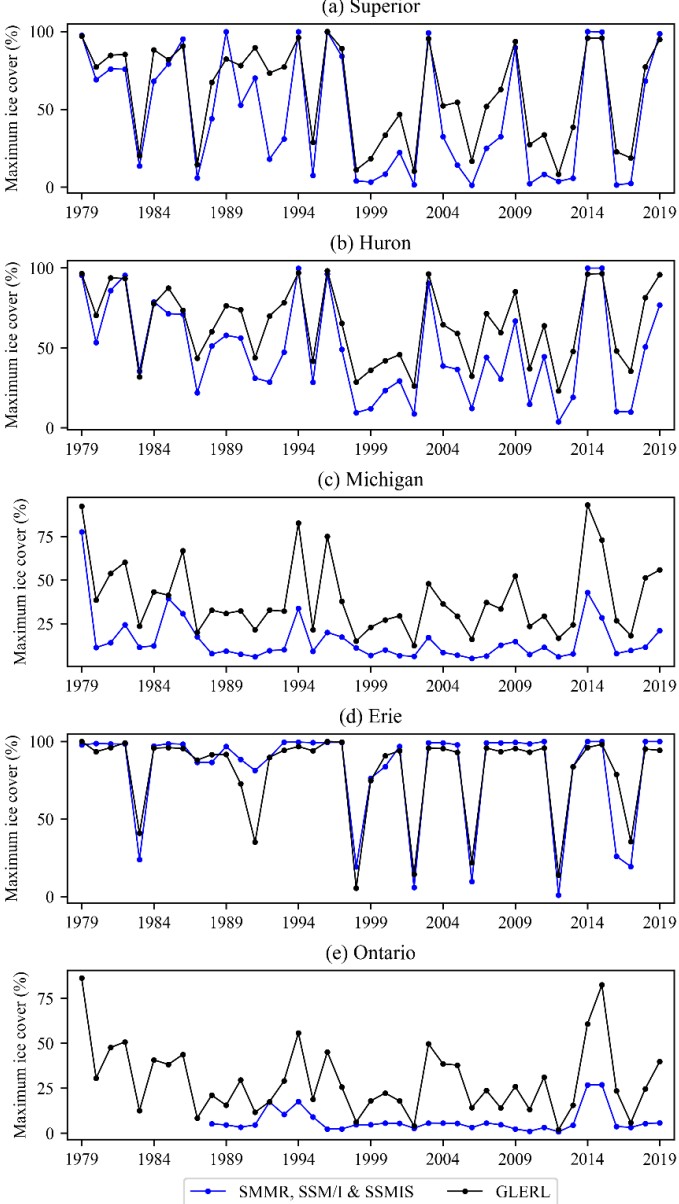

**Figure 7 Comparisons of the annual maximum ice cover (% of lake area) of the Great Lakes from GLERL records and SMMR, SSM/I**
410 **& SSMIS data. (a) Superior, (b) Huron, (c) Michigan, (d) Erie, and (e) Ontario.**

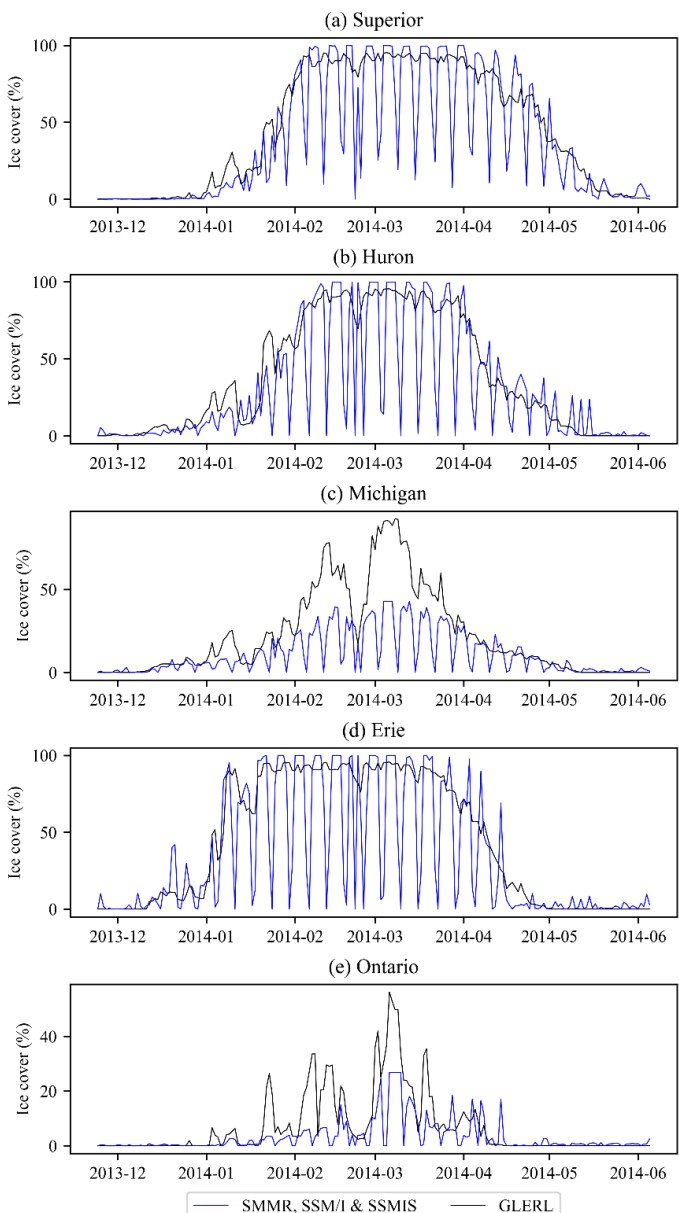

**Figure 8 Comparisons of the daily ice cover (% of lake area) of the Great Lakes from GLERL records and SMMR, SSM/I & SSMIS data in 2014. (a) Superior, (b) Huron, (c) Michigan, (d) Erie, and (e) Ontario.**

### 3.4 Lake ice phenology in the Northern Hemisphere

Among the 56 study lakes, 45 lakes experienced annual ice cover during their entire lake ice phenology records, and the remaining 11 lakes had no ice detected for one year or more. Note that since we only selected the pixels 6.25 km away from the lake shore to extract the lake ice phenology, it is possible that some lakes had ice cover near lake shore, but we did not obtain the information. The average date/days of ice dates and durations for each lake during their entire lake ice phenology records were calculated (relative to 1 September) and are shown in Figure 9. In order to avoid bias in the statistics for the lakes with no ice detected in some years, only the 45 lakes with annual ice cover were included. Two of the 45 lakes (Caspian Sea and Ladoga) did not have annual complete ice cover in winter, hence only their freeze-up start dates, break-up end dates, and ice cover durations were calculated (Figure 9). In addition, the statistical results of ice phenology for the 45 lakes are shown in Table 6, including their

average, medium, minimum (earliest/shortest), and maximum (latest/longest) date/days, as well as the extreme difference (maximum – minimum) and standard deviation. Here we can see that the period of ice formation is from the end of October to January of the next year, and from mid-March to late July for break-up. Overall, the differences among freeze-up dates are smaller than those of break-up dates (with smaller extreme differences and standard deviations, Table 6), but the spatial characteristics of break-up dates are more consistent with latitude (Figure 9). Different lakes experienced a complete freezing duration ranging from 58 to 268 days (average 153 days) and an ice cover duration ranging from 62 to 275 days (average 161 days). Lakes in low-latitude Eurasia had the shortest ice cover durations, while lakes in northern Canada had the longest ice cover durations (Figure 9).

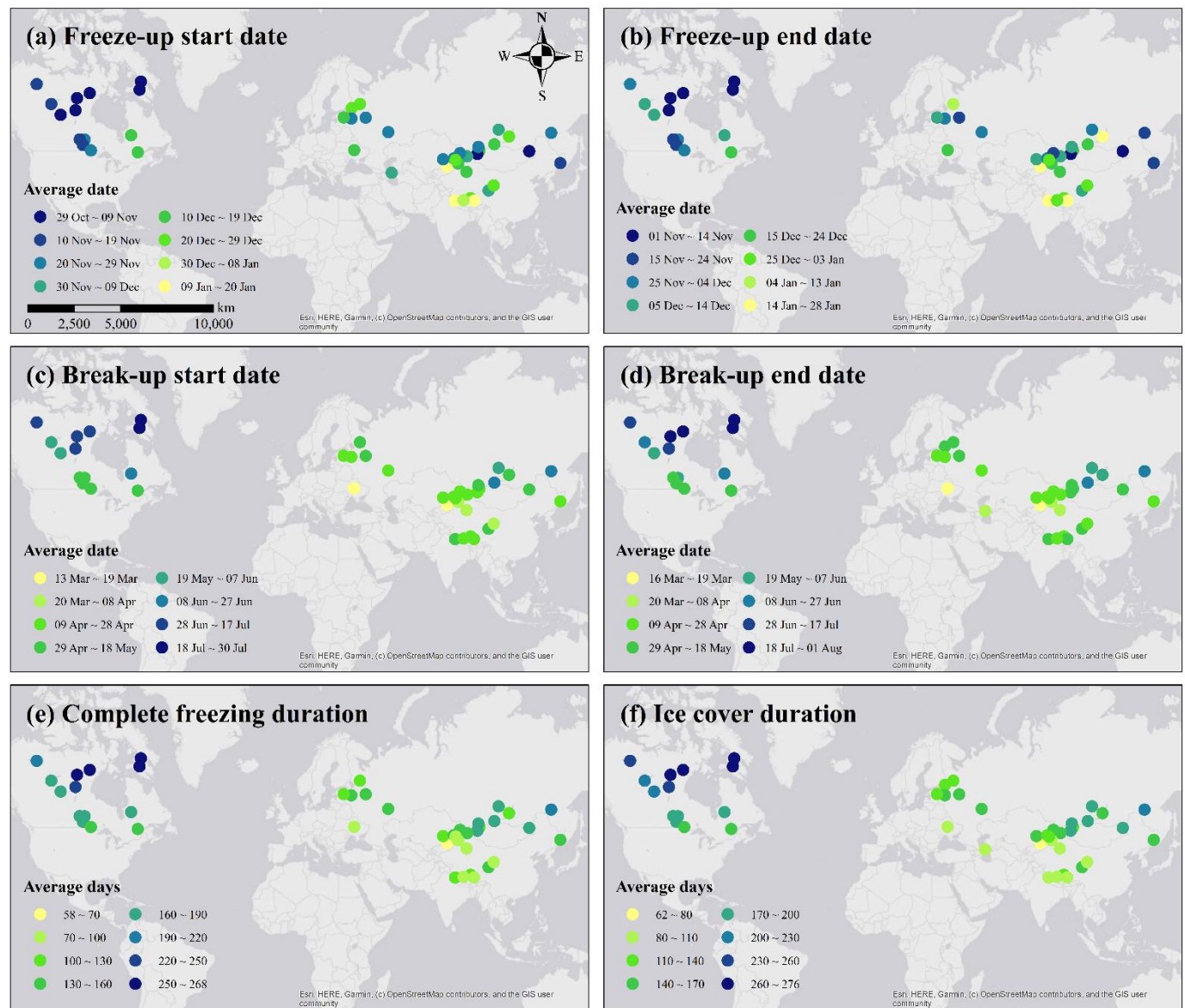

**Figure 9 Average date/days of ice phenology for the study lakes for their entire lake ice phenology records extracted from SMMR, SSM/I & SSMIS data. (a) Freeze-up start date, (b) freeze-up end date, (c) break-up start date, (d) break-up end dates, (e) complete freezing duration, and (f) ice cover duration.**

**Table 6 Average, median, minimum, maximum, extreme difference (Max – Min), and standard deviation (Std.) values of lake ice phenology in the Northern Hemisphere. FUS, FUE, BUS, BUE, CFD, and ICD represent freeze-up start, freeze-up end, break-up start, break-up end date, complete freezing duration, and ice cover duration, respectively.**

|  | Average | Median | Minimum | Maximum | Max-Min | Std. |
|---|---|---|---|---|---|---|
| FUS | 1-Dec | 28-Nov | 29-Oct (Amadjuak & Netilling) | 19-Jan (Nam) | 83 | 22 |
| FUE | 6-Dec | 4-Dec | 31-Oct (Dubawnt) | 27-Jan (Nam) | 88 | 23 |
| BUS | 8-May | 30-Apr | 13-Mar (Qapshaghay Bogeni Reservoir) | 29-Jul (Netilling) | 137 | 35 |
| BUE | 11-May | 2-May | 16-Mar (Qapshaghay Bogeni Reservoir) | 31-Jul (Netilling) | 138 | 35 |
| CFD | 153 | 152 | 58 (Qapshaghay Bogeni Reservoir) | 268 (Netilling) | 209 | 51 |
| ICD | 161 | 156 | 62 (Qapshaghay Bogeni Reservoir) | 275 (Netilling) | 213 | 51 |

## 4 Data availability

The annual lake ice phenology records for the 56 study lakes are available at https://doi.org/10.1594/PANGAEA.937904 (Cai et al., 2021). Calendar dates of freeze-up start, freeze-up end, break-up start, and break-up end, as well as the days of complete freezing duration and ice cover duration are provided in the dataset. Additional data of annual maximum ice cover for the Great Lakes, Caspian Sea, and Ladoga are also provided. Since the original data (CETB dataset) was updated to the end of 2019, we currently provide the lake ice phenology from 1979 to 2019.

## 5 Conclusions

This study used passive microwave SMMR, SSM/I & SSMIS data available on a 3.125 km grid from the CETB dataset to extract the ice phenology for 56 lakes in the Northern Hemisphere from 1979 to 2019. An automatic threshold algorithm based on the MTT method was applied to determine the lake ice status for each pixel, and a buffer of 6.25 km was set to exclude pixels with high land contamination. Then, ice phenology dates were extracted for each lake by the thresholds of 5% and 95% of the total pixels. To keep the lake ice phenology consistent and comparable among different years and different lakes, for the overlapping lake ice phenology results extracted from multiple satellites, we prioritized the results from the satellite with the highest percentage of effective observations.

The main error sources of the lake ice phenology extracted from SMMR, SSM/I & SMMIS data were attributed to the periodically missing data at middle and low latitudes caused by the polar orbit operation mode and the mixed pixels caused by the original coarse spatial resolution of the satellite acquisitions. Ice phenology results for the lakes at low latitudes and/or with small areas tend to be characterized by a larger uncertainty. The switch between sensors and satellites over the time series also resulted in certain differences in the lake ice phenology results, with a bias ranging from -1.30 to 2.00 days and an MAE from 1.20 to 3.13 days.

The lake ice phenology results extracted from SMMR, SSM/I & SSMIS data were compared to the ice dates from in-situ observations and an AMSR-E/2-derived lake ice phenology product. Compared to in-situ observations, the ice-on dates for 4 out of 8 sites, and the ice off dates for 8 out of 10 sites were significantly consistent. The differences between the in-situ observations and the lake ice phenology extracted in this study were mainly due to the different fields of view of human observers versus satellite instruments. As for AMSR-E/2 product, the ice dates showed strong agreement with biases ranging from -1.98 to 2.98 days and MAEs ranging from 2.21 to 3.60 days. The different spatial resolution and sensor characteristics of AMSR-E/2 and SMMR, SSM/I

& SSMIS data can explain the biases between the ice dates. Annual maximum ice cover determined for the Great Lakes also showed significant consistencies compared to GLERL historical ice cover records.

From 1979 to 2019, the average complete freezing duration and ice cover duration for all lakes forming an annual ice cover were 153 days (58 – 268 days for individual lake) and 161 days (62 – 275 days). Lakes in low latitudes of Central Asia had the shortest ice durations, while lakes in northern Canada had the longest ice durations.

The new dataset consists of lake ice phenology records derived from SMMR, SSM/I & SSMIS data from 1979 to 2019. Lake ice phenology (freeze-up, break-up and ice cover duration) is a robust indicator of climate change. The analysis-ready dataset is available to the science/user community to investigate, among several possible topics: 1) trends and inter-annual variability in lake ice phenology across the Northern Hemisphere in response to climate change and atmospheric teleconnections patterns; and 2) the impact of changing ice phenology on local/regional weather, climate and hydrology, aquatic ecosystems, cultural and socio-economic activities.

Finally, regular updates of the lake ice phenology data record are planned with future releases of the CETB dataset. Work is also underway on further improving the lake ice phenology retrieval algorithm and product, and the possibly of adding more lakes in a future release.

## Author contributions

YC and CD conceived the idea. YC designed the code and carried out the data processing. YC prepared the manuscript with contributions from all co-authors. C-QK supervised the study.

## Competing interests

The authors declare that they have no conflict of interest.

## Acknowledgements

This work was supported financially by the National Natural Science Foundation of China (No. 41830105 and No. 42011530120) to Chang-Qing Ke, the program of China Scholarship Council (CSC, No. 201906190109) to Yu Cai and the Natural Sciences and Engineering Research Council of Canada (NSERC, RGPIN-2017-05049) to Claude Duguay. The CETB dataset, GLRIPD data, and AMSR-E/2 lake ice phenology product were obtained from the U.S. National Snow and Ice Data Center (http://nsidc.org/). The GLERL ice cover data were obtained from National Oceanic and Atmospheric Administration (https://www.glerl.noaa.gov/data/ice/#historical). The lake boundaries were obtained from the European Space Agency (ESA) Climate Change Initiative Lakes project (https://climate.esa.int/en/projects/lakes/). The physical characteristics of the study lakes were obtained from the HydroLAKES dataset (https://www.hydrosheds.org/page/hydrolakes)

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
