# Peer review of "A 41-year (1979-2019) passive microwave derived lake ice phenology data record of the Northern Hemisphere"

_Earth System Science Data, 2021_

## Author Comment (AC2)

**Responses to RC1:**

**Summary**

This manuscript presents a new dataset of lake ice phenology for several dozen large lakes in the Northern Hemisphere based on passive microwave data. This dataset, based on a new passive microwave dataset beginning in 1979, represents the most comprehensive passive microwave lake ice phenology dataset to date. The dataset is compared against in situ observations for several lakes, including the Laurentian great lakes, and another passive microwave dataset from the AMSR-E and AMSR-2 instruments and generally found to be consistent with these other datasets, though with greater consistency for larger lakes at higher latitudes.

**Response:** We thank you for your positive comments and constructive suggestions. We think all the comments can be addressed in the revised manuscript. Our responses to each comment are presented as follows.

**Overall Review**

Fundamentally, the dataset presented in this paper is likely to be useful to the scientific community. I recommend some alterations to the paper and to the dataset itself, which I believe will improve it. However, I believe it should likely be published after consideration of reviewer comments. The strengths of the dataset are strongly related to the strengths of passive microwave remote sensing in general. There is a long record of global data, it is not impeded by clouds or other atmospheric effects, and there is a considerable literature suggesting that it can be used to detect ice status in large lakes. The dataset presented here is quite clearly the most comprehensive passive microwave lake ice phenology dataset, and it is likely to be of use to researchers interested in ice phenology patterns, including as a point of comparison for datasets collected using other methods (e.g. ground-based surveys, optical remote sensing, active-source radar).

There are 2 primary weaknesses that I believe should be addressed before the dataset and paper are finally published:

1. The authors refer multiple times to uncertainty in ice phenology for various lakes, but there is no uncertainty field included in the dataset itself for any of the data fields. It is probably impossible to include all sources of uncertainty, but that should not stop the authors from include those sources they can quantify. I have some experience in this area with optical datasets, and what we have generally included are gaps between viable observations (see, for example, Pavelsky and Smith, 2004 and Zhang et al., 2021). These gaps are also clearly present in the passive microwave data (though for different reasons) and are discussed throughout the paper. The authors should at least be able to represent this source of uncertainty. However, they may also be able to represent other sources related to lake size. For example, there are several lakes that are represented by only one

passive microwave pixel. In these cases, ice flagging is binary and likely to be less accurate. In any case, I would like to suggest that the authors quantify uncertainty on all dates as fully as possible and include those estimates in the dataset. Of course this also entails including a description in the paper of what sources of uncertainty are included in the flags.

**Response:** Thank you for the comment. As suggested, we have now performed a quantitative analysis of uncertainty arising from data gap and the representativeness of passive microwave pixels. Characterization of uncertainty will indeed be useful for users interested in using our dataset for climate studies.

We estimated the uncertainty caused by missing data for lake ice phenology dates. The average uncertainty of all records in the dataset were -1 day, of which 61.70% of the records were not affected by missing data. We calculated the average uncertainty for each year and each lake and show the results in Figure R1. The lower frequency of SMMR data led to larger uncertainty (average -3 days) in the lake ice phenology results, which decreased significantly after 1988 (Figure R1a). Figure R1b shows the absolute average uncertainty for each lake ordered by latitude (records extracted from SMMR data were not included because not all lakes had records from 1979 to 1987), it can be seen that the average uncertainty is larger for lakes found at latitudes below 40°N (Ayakkum, Qinghai, Ngoring, Siling, Tangra, Zhari Namco, Nam, and Ma-p'ang yung-ts'o, all on the Tibetan Plateau). In addition, the uncertainty describes the extreme error that may exist in lake ice phenology dates due to missing data, while the actual error is more likely to be smaller than the uncertainty. For example, the F08 data had 43 consecutive days of missing data from 1 December 1987 to 12 January 1988, while Lake Ladoga started to form ice cover during the period and had an ice coverage of 34% on 13 January 1988. As a result, we recorded 13 January 1988 as the freeze-up start date of Lake Ladoga with an uncertainty of -43 days (one of the largest uncertainties), but its actual error was probably much smaller than 43 days.

[Figure]

**Figure R1 Differences in absolute average uncertainty among years and lakes. (a) Absolute average uncertainty for each year; and (b) absolute average uncertainty of lake ice phenology results extracted from SSM/I and SSMIS data for each lake ordered by latitude.**

Before extracting the lake ice phenology dates, we set a buffer of two pixels (6.25 km) to exclude pixels near lake shore. The setting of a buffer will cause the loss of TB information near lake shore. Based on the number of pixels we used and the lake area, we calculated the representativeness of the pixels for each lake. Depending on the lake area and shore complexity, and the possible existence of islands on the lake, the representativeness of the pixels ranged from 0.4% (La Grande 3 Reservoir) to 88.5% (Caspian Sea). For lakes with a low representativeness, the setting of the buffer may result in a non-negligible error in the lake ice phenology results which is hard to quantify. Since the freeze-up and break-up of ice cover usually starts from lake shore (especially for the freeze-up), the beginning signals of freeze-up and break-up extracted from the retained pixels may be later compared with the actual ice conditions, while the ending signals may be earlier.

We will add the statements in the revised manuscript. We are currently preparing a second version of this dataset where the lake ice phenology results will be updated to 2021, and the uncertainties caused by missing data will be added to the dataset. The uncertainty caused by the representativeness of passive microwave pixels which was related to the lake size and shape will be added in the revised manuscript as a table.

2. There are a number of places in the paper where statements are made that may be true but which are not supported by any data or analysis in the paper. These include:

   1. Line 308: "The main reason for the difference between lake ice dates from in-situ observations and passive microwave is their different observation ranges. In-situ records rely on observations of lake ice status visible from lake shores by human observers, while passive microwave satellites record TB from the entire lake surface (here within the pre-defined buffer)." While this seems reasonable to me, there is no evidence in the paper that it is true, and no other work is cited.

   **Response:** Thank you for the comment. As you indicate, while this is a plausible explanation, we did not provide evidence. We will revise the sentence to indicate that the difference is "hypothesized to be related" or "tentatively attributed" to their different observation ranges. We will then add a sentence just after to the effect that a follow-up investigation is indeed needed to quantify differences between in-situ observations with satellite-derived time series. To the best of our knowledge, this has yet to be done and, most importantly, if both sources show similar trends and variability over overlapping time periods.

2. Line 336: "Therefore, AMSR-E/2 data can capture more information near the lake shore than SMMR, SSM/I & SSMIS data, which led to the directional differences between the lake ice phenology dates extracted from the two datasets." Same as above.

**Response:** We also calculated the representativeness of AMSR-E/2 data. The representativeness of AMSR-E/2 data after setting a buffer of 5 km ranged from 1.9% to 89.8%, with an average of 5.0% higher than that of SMMR, SSM/I & SSMIS data. We will add the explanation in the revised manuscript.

3. Line 355: "The correlations between the ice cover for Huron and Ontario were lower than that of the other three lakes, which is also because the ice cover of these two lakes were usually small, and the ice first forms near shore which may not be covered in the set buffer (Figure 5)." This assessment potentially conflicts with the assessment shown in the next paragraph indicating that problems with the MTT algorithm my result in lower accuracies for Lake Ontario.

**Response:** We will rewrite this section to explain the differences in ice cover maximums from SMMR, SSM/I & SSMIS data and GLERL. The description of the uncertainty caused by the MTT algorithm will be moved to the Uncertainty Analysis section, and two paragraphs in this section will be merged. The MTT algorithm may be insensitive to short-term ice cover or frequent melt-refreeze event during the break-up or freeze-up seasons due to the application of smoothing approaches, which also result in the underestimation of ice coverage in winter. The lower accuracies for Michigan and Ontario will be pointed out as an example to explain how short-term, large-area ice coverage failed to be detected by the algorithm.

4. Line 435: "The differences between the in-situ observations and the lake ice phenology extracted in this study were mainly due to the different fields of view of human observers versus satellite instruments." Again, while this is plausible, no evidence of this explanation is explicitly shown in the paper.

**Response:** See our reply to first comment above.

I would strongly recommend that the authors either qualify these statements, provide evidence for them, or remove them. I would tend to hope for one of the two former options, as long as more concrete evidence suggests that they are correct.

Other than these two substantial areas for potential improvement, all other comments (listed below) are minor.

**Specific Points**

Line 14: I would write ". . . in middle and high latitude regions."

**Response:** We will modify it in the revised manuscript.

Line 17: An alternate to what?

**Response:** It is an alternate source to other satellite data that can provide daily observations, such as optical, active microwave, and raw passive microwave data. We will add it in the revised manuscript.

Line 28: I would write "consistency" rather than "consistencies" in both cases in this line.

**Response:** We will replace them.

Lines 39-40: I would recommend citing a few papers here, as there are a number of papers that have looked at this. For example, Smejkalova et al., 2016; Sharma et al., 2016.

**Response:** We will add the references.

Line 42: Might also consider citing Knoll et al., 2019 here.

**Response:** We will add the reference.

Line 50: I might change the wording here, as a reader could be confused into thinking that all 865 site records begin in 1443, when most of them begin much latter.

**Response:** We agree with your suggestion, we will change the sentence to "the Global Lake and River Ice Phenology Database (GLRIPD) from the National Snow and Ice Data Center (NSIDC) contains ice phenology records for 865 sites, including 24,438 ice-on records and 33,370 ice-off records, where the earliest record can date back to 1443" in the revised manuscript.

Line 72: I might, again, cite the Smejkalova et al. 2016 paper here.

**Response:** We will add the reference.

Line 101: I would probably delete "Satellite" before Nimbus-7.

**Response:** We will delete it.

Line 104: I would replace "data of the" with "data from the"

**Response:** We will modify it.

Line 108: "is against" should be "is compared against" or similar.

**Response:** We will modify it.

Figure 1: perhaps it's unnecessary to do, but I believe there are a number of Patagonian lakes that might be close to the necessary size threshold for inclusion in this study. If the authors have not looked at these lakes to see if they are viable, I would recommend doing so given the paucity of similar records in the southern hemisphere.

**Response:** Since this paper focuses on the Northern Hemisphere, we did not contain lakes in the Southern Hemisphere. According to the new ESA CCI+ lakes dataset (v2.0 and 2.01 product), very few lakes formed ice in the Southern Hemisphere (Figure R2). We used three years of F14 data to detect ice status for 47

lakes in the Southern Hemisphere with at least one pixel 6.25 km away from the lake shore, and found that none of the lakes had ice cover detected by the algorithm. We also checked optical images for Argentino Lake, Viedma Lake, and San Martin Lake over the past two years and found that they were not covered by lake ice in winter. Probably these lakes freeze only in very cold years.

[Figure]

**Figure R2 Lake ice cover flag from the new ESA CCI+ lakes dataset. Magenta means the lake does not form ice, and blue means the lake forms ice.**

Line 141: Typo at the end of the line. Should be "For comparison with lake ice phenology. . ."

**Response:** We will modify it.

Line 144: I would write "on 7 lakes" instead of "of 7 lakes."

**Response:** We will modify it.

Line 149: I would write "from the CETB dataset."

**Response:** We will modify it.

Line 155: I would write "from the National. . ."

**Response:** We will modify it.

Line 184: I would write "For the remaining pixels. . ."

**Response:** We will modify it.

Line 185: Why were thresholds of 5% and 95% chosen? Is there any sensitivity compared to say, 80-90% and 10-20%?

**Response:** We believe that as long as the change curve of the number of lake ice pixels is clear, the lower the threshold means the smaller the error, so we chose 5% and 95%, which showed high applicability in the extraction of lake ice phenology. As we mentioned in the manuscript, 97.92% of all the records were

successfully extracted by the thresholds, but still some records were extracted by looser thresholds. To increase the completeness of the lake ice phenology results, the use of thresholds was relatively flexible.

Line 194: This choice to omit ice covered periods of <30 days is significant. Would you then say that your estimates of ice duration likely underestimate total ice duration, as they ignore any intermittent ice cover occurring during the breakup or freezeup seasons? If so, I would explicitly mention this.

**Response:** We agree with this comment. We only kept the lake ice phenology results with ice cover persisting for more than 30 consecutive days, which may also lead to some lakes with short-term ice cover being recorded as ice-free. Not only this step, the smoothing algorithm and the setting of the buffer may also lead to an underestimation of the ice cover duration, especially for small lakes with irregular shape and short-term ice cover. We will add the statements in the revised manuscript.

Line 228: I would write "For years with multiple lake ice records. . ."

**Response:** We will modify it.

Lines 246-249: A minor point, but given the size of these lakes, I'm not sure there's any need to include the fourth significant digit in latitude. All are nearly 0.5 degree (or more) in N-S extent.

**Response:** We agree with your suggestion, we will modify it in the revised manuscript.

Line 270: I would recommend making sure you stay in consistently the present or past tense for this sentence and the following one.

**Response:** We will modify it.

Line 276: I would write "periodically missing data" instead of "periodical data missing."

**Response:** We will modify it.

Line 301: I would write "compared" rather than "used to compare"

**Response:** We will modify it.

Line 302: I would write "complete ice cover" rather than "completely ice covered."

**Response:** We will modify it.

Lines 313-316: while I agree with the statement in this sentence to a degree, I would argue that the agreement with GLRIPD records is not particularly strong in many cases. As such, I'm not sure I agree that the analysis presented here provides strong evidence for this statement. Rather, I would say that remotely sensed observations can complement in situ measurements.

**Response:** We agree with this comment. We will mention the complementarity between remotely sensed and in situ observations in the revised manuscript. With the decrease in in-situ observation sites globally since the 1980s, there has been a shift towards the increased use of remote sensing technology for lake ice monitoring (Murfitt and Duguay, 2021).

Lines 335-336: Make sure to keep verb tenses the same in this sentence.

**Response:** We will modify it.

Line 429: The conclusion that lakes at low latitudes and/or small areas tend to have larger uncertainties would be much more robust if there were a more consistent uncertainty quantification, as mentioned above.

**Response:** We agree with this comment. Since we will add quantitative analysis for uncertainties arising from data gap and the representativeness of passive microwave pixels in the Uncertainty Analysis section, we believe this conclusion will be more convincing in the revised manuscript.

---

## Author Comment (AC3)

**Responses to RC2:**

Overall review

The authors have used an automated method to extract ice phenology data from passive microwave data. The data set presented here and explained in the article is generally very interesting and will be useful for research community. The data set is likely the longest and most comprehensive ice phenology data set from satellite-based observations for that large number of lakes. This data covers multiple climatological areas and lake sizes and is therefore well worth publication. Data set is usable in the present format.

**Response:** We thank you for your positive comments and constructive suggestions. We think all the comments can be addressed in the revised manuscript. Our responses to each comment are presented as follows.

Comments:

I would like to present 2 recommendations to improve the usability of the data and the manuscript.

Data set does not include any sort of error estimates for dates, duration, or maximum ice cover area. In the manuscript is long discussion on the errors and their possible sources, but these should be quantified in the data, or at least in the manuscript. It is very difficult to compare this data set to other similar data sets without this information. In the manuscript one major target for this data is climate research, it is difficult to draw conclusion if error marginals are unknown. To use this data to complement data gaps of in situ archives of ice phenology, more precise definition of the errors and their sources compared to the GRLIPD and GLERL ice cover data sets should be included.

**Response:** Thank you for the comments. Since Reviewer 1 also mentioned the necessity of quantitative uncertainty analysis, we collected the uncertainty caused by missing data for lake ice phenology dates and calculated the representativeness of passive microwave pixels relative to lake area. The average uncertainty of all the records in the dataset were -1 day, of which 61.70% of the records were not affected by missing data. Overall, the uncertainty of results after 1992 was much smaller than before, and the uncertainty for lakes located at a latitude below 40°N was relatively larger. The representativeness of the pixels ranged from 0.4% (La

Grande 3 Reservoir) to 88.5% (Caspian Sea) depending on the lake area and shore complexity, and the possible existence of islands on the lake. We will add the quantitative analysis in the revised manuscript, the uncertainties caused by missing data will be added to the dataset and the uncertainty caused by the representativeness of passive microwave pixels which was related to the lake size and shape will be added in the revised manuscript as a table.

Using 37 GHz H-polarized data has some limitations in distinguishing ice and open water. Signal can be strongly affected by open water surface roughness from wind (for example, K.-K. Kang et al.: Estimating ice phenology on large northern lakes from AMSR-E; doi:10.5194/tc-6-235-2012). This problem and its implications to the data is not discussed in the manuscript at all, and it is not covered in any of the references provided. By discussing this matter or providing references that discuss this, will make this data much more reliable and usable.

**Response:** We agree with that 37 GHz H-polarized is sensitive to wind-induced surface roughness during the open water period. Du et al. (2017) mentioned that although microwave emissions from a lake are determined by many factors including the surface roughness, sharp changes in TB observations at multiple frequencies are evident during the transitions between lake freeze-up and breakup periods. Moreover, the effect of wind-induced surface roughness can be attenuated by the smoothing approaches in MTT algorithm. But it may still lead to errors in lake ice phenology results. While the study of Du et al. (2017) clearly demonstrated the strength of the MTT approach applied to 37 GHz data, we will still add this to the discussion in the revised manuscript.

I also have some minor comments to consider:

on line 176: "When the lake is water covered, the TB for land-contaminated pixels will be higher than that of a pure pixel, while when the lake is ice covered, the TB will be lower than that of pure pixel." Last 2 words: Is it pure pixel of ice/water/land?

**Response:** When the lake is water covered, the TB for land-contaminated pixels will be higher than that of a pure water pixel, while when the lake is ice covered, the TB will be lower than that of a pure ice pixel. We will modify it in the revised manuscript.

on line 271: "When the lake area was large enough, the gradual freeze-up or break-up within

the pixel can be ignored, but for small lakes, it may lead to certain deviations in the lake ice phenology results." What are the certain deviations?

**Response:** Before the TB exceeds (/falls below) the threshold, the lake surface within the pixel may have already started to freeze-up (/break-up), and this process may not end even after we detect the ice covered (/water covered) signal. As a result, the beginning signals of freeze-up and break-up extracted from retained pixels may be later compared with the actual ice conditions, while the ending signals may be earlier. We will add the explanation in the revised manuscript.

on line 312: "Overall, if the overlapping time between the two dataset was longer, the lake ice dates could show a higher consistency." How or Why that could be the case?

**Response:** We will delete the statement in the revised manuscript. The difference between lake ice dates from in-situ observations and passive microwave is tentatively attributed to their different observation ranges, but a follow-up investigation is still needed to quantify and explain the differences between in-situ observations with satellite-derived time series.

on line 353:" This is because a buffer of 6.25 km was used to exclude pixels near the lake shore, which happens to be the place where lake ice forms first." If this is the only explanation in the difference between GLERL data and this data set, one would expect the difference to gradually wannish as one nears 100% ice coverage. This is not the case in all the lakes in all the years. Why is that?

**Response:** The limitation of MTT algorithm's insensitivity to short-term ice cover would also lead to difference between GLERL data and the ice cover extracted from SMMR, SSM/I & SSMIS data. And this is also why sometimes the lake has been 100% ice covered but only partial coverage was detected by SMMR, SSM/I & SSMIS data. We will modify the explanation in the revised manuscript.

---

## Author Response (AR2)

**Responses to Referee #3:**

This study developed an approach to monitor lake ice phenology based on passive microwave brightness temperature (TB) products. By applying this method to 56 lakes from 1979 to 2019, a ice phenology dataset was established. Considering the benefits of passive microwave data including long record, daily re-visit frequency and potential to work under all-weather conditions, this study is of great interest and meaningful for the community to improve our understanding of global lake ice phenology. The paper is well written and the methodology flow is clear. Therefore, I recommend accepting this paper after minor revisions.

**Response:** Thanks for your recommendation.

1.Line 97, compared with the traditional studies that only a few large lakes could be monitored based on passive microwave data, more lakes with smaller surface areas are included study because the latest TB product with higher spatial resolution was released by NSIDC. I suggest giving more background information about this new TB product, such as how it was achieved and what the accuracy of this down scaled product.

**Response:** We have added some brief introductions to the CETB dataset here (see Lines 99-101 in the revised manuscript): "The enhanced-resolution images were generated using the radiometer version of the Scatterometer image Reconstruction (rSIR) algorithm, which provides higher spatial resolution surface $T_B$ images with smaller total error compared with conventional drop-in-the-bucket gridded image formation (Long and Brodzik, 2016)." There are also additional introductions to the dataset in the data description section.

2.Methodology section in 2.3.1

I think add a figure showing the time series of TB for one or two lakes would be better to illustrate how MIT works for determining the abrupt changes.

**Response:** Thanks for the suggestion. We have added an example of Great Bear Lake to illustrate how to determine lake ice status for a pixel and extract ice phenology for a lake in the revised manuscript (Figure R1).

[Figure]

**Figure R1** Example of determining the lake ice status for a pixel and extracting the ice phenology for a lake. (a) The $T_B$ image of Great Bear Lake on 1 July 2014, (b) variations in the $T_B$ of the pixel marked in red in (a), the red line represents the reference $T_B$ determined by MTT algorithm, and the pixel was determined to be water covered on 1 July 2014 (red dot), (c) lake ice status for all the pixels at least 6.25 km away from the lake shore on 1 July 2014, and (d) variations in the number of lake ice and open water pixels for Great Bear Lake in 2014, the lake ice phenology were extracted by thresholds of 5% and 95% of the total lake pixels, and 1 July 2014 (red dot on the ice cover line) was determined to be in break-up period.